# Flow Field Reconstruction with Sensor Placement Policy Learning

**Ruoyan Li**[1], **Guancheng Wan**[1], **Zijie Huang**[1], **Zixiao Liu**[1],
**Haixin Wang**[1], **Xiao Luo**[1], **Wei Wang**[1], **Yizhou Sun**[1]
[1]University of California, Los Angeles

## Abstract

Flow-field reconstruction from sparse sensor measurements remains a central challenge in modern fluid dynamics, as the need for high-fidelity data often conflicts with practical limits on sensor deployment. On one hand, existing deep learning–based methods have demonstrated promising results, but they typically rely on overly simplified assumptions such as 2D domains, predefined governing equations, synthetic datasets derived from idealized flow physics, and unconstrained sensor placement. In this work, we address these limitations by studying flow reconstruction under realistic conditions and introducing a *directional transport-aware Graph Neural Network (GNN)* that explicitly encodes both flow directionality and information transport. On the other hand, conventional sensor placement strategies frequently yield suboptimal configurations. To overcome this, we propose a novel *Two-Step Constrained PPO* procedure for Proximal Policy Optimization (PPO), which jointly optimizes sensor layouts by incorporating flow variability and accounts for reconstruction model's performance disparity with respect to sensor placement. We conduct comprehensive experiments under realistic assumptions to benchmark the performance of our reconstruction model and sensor placement policy. Together, they achieve significant improvements over existing methods.

## 1 Introduction

Flow field reconstruction from sparse sensor data (Berkooz et al., 1993; Schmid, 2010) has emerged as a pivotal challenge in modern fluid dynamics, particularly as the demand for high-fidelity measurements clashes with the practical constraints of sensor deployment. Such reconstruction techniques underpin real-world applications such as aerodynamic shape optimization and active flow control in aerospace and turbomachinery (Luo et al., 2017). With the rapid advancement of deep learning, leveraging deep learning models to transform limited experimental data into detailed, reliable representations of complex flow phenomena has been a promising solution (Zhong et al., 2023; Yadav et al., 2025; Xu et al., 2023; Jing et al., 2024; Li et al., 2025).

On one hand, we note that many existing studies rely on assumptions that may not hold in realistic scenarios. Specifically, these works commonly assume that: (1) **Domain**: Experiments are predominantly conducted in two-dimensional (2D) domains. However, real-world applications take place in three-dimensional settings. (2) **Physics**: The governing physical PDEs, such as the Navier–Stokes equations, are known a priori, which is integrated to inform and constrain the models. Yet, empirical fluid dynamic data rarely conform precisely to the Navier–Stokes equations (Hadjiconstantinou, 2006; Stubbe, 2020). (3) **Datasets**: Datasets are usually generated through pseudo-spectral solvers (Orszag, 1969) or reynolds-averaged Navier-Stokes (RANS) (Tennekes & Lumley, 1992), but these numerical solvers rely on simplified assumptions about fluid behavior, yielding datasets that diverge from real-world fluid dynamics. (4) **Sensor Placement**: Sensors are assumed arbitrarily placed within the flow field without influencing the fluid dynamics. In reality, measurement sensors should either

39th Conference on Neural Information Processing Systems (NeurIPS 2025).

Table 1: Comparison of related works on problem assumptions.

| Paper | 3D-Domain | Unknown Physics | Complex Data | Placement Optimization |
|---|---|---|---|---|
| Zhong et al. (2023) | ✗ | ✓ | ✗ | ✗ |
| Mo & Magri (2024) | ✗ | ✗ | ✗ | ✗ |
| Yadav et al. (2025) | ✗ | ✗ | ✗ | ✗ |
| Jing et al. (2024) | ✗ | ✗ | ✗ | ✗ |
| He et al. (2022) | ✗ | ✓ | ✗ | ✗ |
| Zhang et al. (2022) | ✓ | ✓ | ✗ | ✗ |
| Hosseini & Shiri (2024) | ✗ | ✗ | ✗ | ✗ |
| Zhang et al. (2025) | ✓ | ✓ | ✗ | ✗ |
| Shan et al. (2024) | ✗ | ✗ | ✗ | ✗ |
| Xu et al. (2023) | ✗ | ✗ | ✗ | ✗ |
| **Ours** | ✓ | ✓ | ✓ | ✓ |

remain fixed at the domain boundaries or be advected with the fluid flow. A comprehensive review of these assumptions is provided in Table 1.

To address this, we generate four three-dimensional (3D) turbulent flow datasets using Direct Numerical Simulation (DNS) in COMSOL (COM, 2020) with varying geometries. The simulations initiate with a randomly generated velocity field, and the inlet velocity is modeled as both time-dependent and stochastic. We argue that training on these datasets enables improved transferability to real-world tasks and yields more reliable evaluation results. Although large-scale, real-world sensor datasets remain unavailable, we propose that the combination of high-fidelity simulations in COMSOL and the incorporation of time-dependent, stochastic inlet conditions provide a more faithful representation of actual fluid phenomena. Further, we restrict sensor placement to the domain boundaries, since permitting sensors to be advected with the fluid flow leads to their accumulation in vortical regions and makes reconstruction of other areas infeasible. **Thus, we aim to develop a reconstruction model that can adapt to arbitrary mesh-based geometries without assuming any underlying PDEs, while confining sensor placement solely to the boundaries.**

We propose a directional transport-aware GNN that explicitly encodes directionality and information transport in the message-passing stage. The explicit parameterization of directional weightings and transported quantities not only mimics the continuous advection operator in a discrete, mesh-based setting but also corresponds to a learnable interpolation algorithm. This encourages learning meaningful representations and yields robust imputation capability across various sensor configurations.

One the other hand, we reveal that traditional methods for sensor placements, such as singular value decomposition (SVD) or QR pivoting (Chmielewski et al., 2002), often perform poorly when integrated with our reconstruction models. We attribute the performance degradation to reconstruction model's performance disparity with respect to sensor placement. **To address this, we train a PPO policy that determines optimal sensor configurations and introduce a novel** *Two Step Constrained PPO* **training procedure to enforce sensor constraints.** Our policy not only captures variability in the fluid field but also accounts for model's performance disparity. Experimental results demonstrate substantial improvements in reconstruction accuracy when using the learned sensor placements.

Our contributions are as follows: **(i) Problem Identification:** We introduce a realistic problem formulation for fluid-field reconstruction and generate extensive datasets that closely mimic real-world scenarios; **(ii) Practical Solution:** To tackle fluid field reconstruction on arbitrary mesh-based geometries, we develop a directional transport-aware GNN that explicitly encodes both directionality and information transport; **(iii) Further Scientific Discoveries:** We find that conventional sensor placement algorithms fail to identify effective sensor locations, and thus we propose a novel *Two-Step Constrained PPO* training strategy to learn a policy that identifies the optimal placement of sensors; **(iv) Experimental Validation:** We conduct comprehensive experiments under realistic assumptions to validate the superiority of our reconstruction model and sensor placement policy.

## 2 Related Work

**AI for Computational Fluid Dynamics (CFD)** Recent advances in machine learning have led to various learning-based surrogate models for accelerating scientific discoveries (Li et al., 2025; Huang et al., 2024b; Wang et al., 2024). In the study of flow field reconstruction, Zhong et al. (2023) leverages a combination of MLP with CNN to reconstruct unsteady vortical flow fields near airfoils.

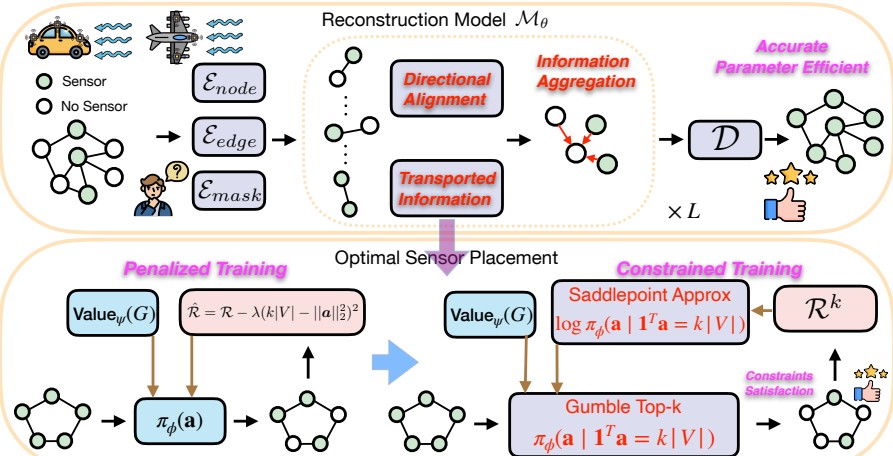

Figure 1: Overall framework of the proposed method. The directional transport-aware reconstruction GNN $\mathcal{M}_\theta$ takes boundary sensor inputs and infers missing field values through message passing with explicit directional alignment and information transport. A two-stage PPO training procedure identifies optimal sensor placements via a penalized stage that softly enforces sensor-count constraints through rewards, followed by a constrained stage where sampling from a constrained probability distribution ensures compliance with sensor limits.

He et al. (2022) introduces the Flow Completion Network, which employs GNNs to reconstruct both structured and unstructured data. Hosseini & Shiri (2024); Shan et al. (2024); Yadav et al. (2025); Xu et al. (2023); Jing et al. (2024) leverages underlying physics or PDE to develop physics-informed neural networks for enhancing reconstruction quality. Mo & Magri (2024) injects artificial noise into sensor data and develops a physics-constrained CNN for reconstruction. For deep learning-based optimal sensor placement, Marcato et al. (2023) employ differentiable programming to integrate sensor placement into the training of a neural network model. Nonetheless, this method is constrained by a fixed number of sensors and makes the assumption that sensors can be positioned arbitrarily without interfering with the fluid dynamics.

# 3    The Fluid Field Reconstruction Model

**Problem Statement** Let $G = (V, E)$ be a graph representing a discretized mesh of the domain boundary, where each vertex $v_i \in V$ corresponds to a node in the mesh and each edge $e_{i,j} \in E$ captures the local connectivity among these nodes. Each node is characterized by a feature vector $v_i = [\boldsymbol{u}_i, \boldsymbol{p}_i, \boldsymbol{a}_i]$, where $\boldsymbol{u}_i \in \mathbb{R}^3$ denotes the velocity, $\boldsymbol{p}_i \in \mathbb{R}$ denotes the pressure, and $\boldsymbol{a}_i \in \{0, 1\}$ is a binary mask indicating the availability of sensor. $\boldsymbol{a}_i = 1$ implies that the velocity and pressure at node $i$ are known (i.e., a sensor is present), while $\boldsymbol{a}_i = 0$ indicates that these values must be imputed and $\boldsymbol{u}_i$ and $\boldsymbol{p}_i$ are randomly generated. Our objective is to develop a reconstruction model $\mathcal{M}_\theta$ that takes the graph $G$ as input and outputs estimated values $[\hat{\boldsymbol{u}}_i, \hat{\boldsymbol{p}}_i]$ for all nodes where $\boldsymbol{a}_i = 0$.

**Directional Transport-Aware GNN** This problem is substantially more complex due to the absence of known governing equations, the use of irregular 3D geometries, and the restriction of sensors to boundaries. Our proposed model is based on an Encoder-Processor-Decoder framework. It is designed to handle these challenges by learning flexible, geometry-aware representations that generalize across diverse domains without relying on explicit physical priors. In the encoding stage, we employ three distinct MLPs denoted by $\mathcal{E}_{mask}, \mathcal{E}_{node}, \mathcal{E}_{edge}$ to embed the node and edge features into latent space. Formally,

$$\tilde{\boldsymbol{a}}_i \leftarrow \mathcal{E}_{mask}(\boldsymbol{a}_i), \quad \tilde{v}_i^0 \leftarrow \mathcal{E}_{node}(\boldsymbol{u}_i || \boldsymbol{p}_i || \tilde{\boldsymbol{a}}_i), \quad \tilde{e}_{i,j}^0 \leftarrow \mathcal{E}_{edge}(e_{i,j}), \tag{1}$$

where $||$ denotes concatenation. Next, we introduce the directional transport-aware processor that integrates the notion of directionality and information transport into the message-passing framework. In our approach, the directional information, $\boldsymbol{d}$, is computed as the inner product between a node's latent representation and its corresponding edge features. This inner product quantifies the degree of alignment of a node with respect to the direction of information transfer, thereby acting as a proxy for the node's contribution to feature reconstruction. The computed directional score is then

used to weigh the differences between the latent states of adjacent nodes, effectively capturing the information transported from node $i$ to node $j$. The resulting directional transport function, $\mathcal{T}$, is parameterized by MLPs, and, subsequently, a node aggregation function, $\mathcal{S}$, synthesizes the weighted edge messages to update the latent state of each node.

$$d_{i,j}^{\ell-1} \leftarrow \langle \tilde{v}_i^{\ell-1}, \tilde{e}_{i,j}^{\ell-1} \rangle, \quad \tilde{e}_{i,j}^{\ell} \leftarrow \mathcal{T}(d_{i,j}^{\ell-1} \cdot (\tilde{v}_j^{\ell-1} - \tilde{v}_i^{\ell-1})), \quad \tilde{v}_i^{\ell} \leftarrow \mathcal{S}(\tilde{v}_i^{\ell-1} || \sum_j \tilde{e}_{i,j}^{\ell}), \quad (2)$$

We apply $L$ layers of this processor with residual connections. Finally, the decoder $\mathcal{D}$ uses one MLP to map each node embedding $v_i^L$ to the desired output space: $[\hat{\boldsymbol{u}}_i, \hat{\boldsymbol{p}}_i] = \mathcal{D}(v_i^L)$.

**Remark 1 (Connection to advection operator)** The design of the processor is closely connected to the advection operator. Advection describes the transport of properties, such as heat or pollutants, via the bulk motion of a fluid. Mathematically, this process is typically characterized by the operator $\boldsymbol{\nu} \cdot \nabla \varphi$, where $\boldsymbol{\nu}$ denotes the velocity and $\varphi$ is the field being transported.

Our $\boldsymbol{d}$ encodes the direction of information flow, acting in the role of $\boldsymbol{\nu}$. The difference between neighboring latent states, $\tilde{v}_j^{\ell-1} - \tilde{v}_i^{\ell-1}$, serves as a discrete analogue to the spatial gradient. By combining these two quantities, we recreate the behavior of the advection operator in high-dimensional latent spaces and thereby enable efficient information propagation within the processor.

**Remark 2 (Connection to interpolation algorithm)** Our processor can be viewed as a learnable interpolation operator. In a generic interpolation scheme, one writes $v_i \leftarrow \sum_{j \in \mathcal{N}(i)} b(v_i, v_j, e_{i,j}) q(v_i, v_j)$, where $v_i$ is the interpolated value, $b(v_i, v_j, e_{i,j})$ is the weight assigned to neighbor $j$, and $q(v_i, v_j)$ is the contribution of node $j$. In our formulation, the weight function is the directional information, $\boldsymbol{d}$, while the neighboring contribution is the difference between the latent states. We also include self-contribution by defining $b(v_i, v_i, e_{i,i}) q(v_i, v_i) = v_i$. Rather than performing a fixed weighted sum, our processor replaces these terms with learnable MLP-based embeddings for both the transported information and the aggregation step.

**Remark 3 (Connection to conventional message passing GNNs)** Compared to conventional message passing GNNs, which typically concatenate node and edge information and process with MLPs, our method explicitly incorporates the directionality of information flow. This explicit encoding facilitates the learning of more meaningful representations, as it encourages the propagation of information along physically motivated pathways.

## 4 Sensor Placement Optimization

Building upon the directional transport-aware reconstruction model described above, we explore methodologies for optimal sensor placement aimed at further enhancing reconstruction accuracy.

**Traditional Approach** We evaluate sensor placements by applying two greedy column-selection strategies: QR pivoting and D-optimality (Manohar et al., 2018). Details on implementation and experimental results are in Appendix G. These methods incur substantially higher reconstruction errors than uniformly distributed sensors. Deep learning models often exhibit performance disparity. Although a sensor placement strategy may be ideal for capturing flow-field variability, our reconstructions model may reconstruct this configuration with lower accuracy than they do for other placements. An effective sensor placement strategy must address both the variability of the fluid field and the reconstruction model's differential subgroup performance. Thus, we aim to train a policy to account for these two factors in sensor placement.

**Problem Statement** Given a reconstruction model $\mathcal{M}_\theta$, we aim to learn a policy $\pi_\phi(\boldsymbol{a}|G)$ that takes a mesh $G$ containing complete fluid field information as input and outputs a probability parameter $\boldsymbol{p}_i \in [0,1]$ of a Bernoulli distribution for each node $i$ for its sensor placement. We interpret a random variable $\boldsymbol{a}_i \sim \text{Bernoulli}(\boldsymbol{p}_i)$ such that $\boldsymbol{a}_i = 1$ if node $i$ has a sensor and $\boldsymbol{a}_i = 0$ otherwise. The policy aims to maximize the reward under such actions, which is the negation of the Mean Squared Error (MSE) of the reconstructed velocity and pressure. The objective function is defined as $\mathcal{R}(\boldsymbol{a}|G) = -\text{MSE}[\{\hat{v}_i \mid \hat{v}_i = \mathcal{M}_\theta(G(\boldsymbol{a})), \boldsymbol{a}_i = 0\}, \{v_i | \boldsymbol{a}_i = 0\}]$. We impose the constraint $\sum_{i=1}^{|V|} \boldsymbol{a}_i = k|V|$, where $|V|$ denotes the cardinality of the set and $k \in [0,1]$ denotes the proportion of the mesh nodes that contains a sensor. Empirical results show that increasing the number of sensors consistently improves reconstruction accuracy. Therefore, instead of imposing an upper bound on sensor count, we enforce the exact number of sensors via this equality constraint. The placement strategy is constrained to assign sensors exclusively to predefined nodes, rather than

---

**Algorithm 1** Two Step Constrained PPO

---

**Require:** Initial policy parameters $\phi_0$, initial value function parameters $\psi_0$
1: # Penalized Training
2: **for** $w = 1, \ldots, T_1$ **do**
3:     Collect $\mathcal{D}_w = \{(G_w, \boldsymbol{a}_w))\}$ by running $\pi_{\phi_w}(\boldsymbol{a} \mid G)$.
4:     Compute penalized reward $\hat{\mathcal{R}}^w = \mathcal{R}^w - \lambda(k|V| - ||\boldsymbol{a}||_2^2)^2$ and advantage estimates $A^w$.
5:     Compute $\log \pi_{\phi_w}(\boldsymbol{a} \mid G)$ and compute ratio $r(\phi)$.
6:     Update policy: $\phi_{w+1} = \arg\max_\phi \frac{1}{|\mathcal{D}_w|} \sum_{G,\boldsymbol{a}} \min\left(r(\phi)A^w, \text{clip}\left(r(\phi), 1-\epsilon, 1+\epsilon\right)A^w\right)$.
7:     Fit value function $\psi_{w+1} = \arg\min_\psi \frac{1}{|\mathcal{D}_w|} \sum_{G,\boldsymbol{a}} \left(\text{Value}_\psi(G) - \hat{\mathcal{R}}^w\right)^2$.
8: **end for**
9: # Constrained Training
10: **for** $w = T_1, \ldots, T_2$ **do**
11:     Collect $\mathcal{D}_w = \{(G_w, \boldsymbol{a}_w))\}$ by running constrained $\pi_{\phi_w}(\boldsymbol{a} \mid G, \mathbf{1}^T\boldsymbol{a} = k|V|)$.
12:     Compute reward $\mathcal{R}^w$ and advantage estimates $A^w$.
13:     Estimate $\log \pi_{\phi_w}(\boldsymbol{a} \mid G, \mathbf{1}^T\boldsymbol{a} = k|V|)$ and compute ratio $r(\phi)$.
14:     Update policy: $\phi_{w+1} = \arg\max_\phi \frac{1}{|\mathcal{D}_w|} \sum_{G,\boldsymbol{a}} \min\left(r(\phi)A^w, \text{clip}\left(r(\phi), 1-\epsilon, 1+\epsilon\right)A^w\right)$.
15:     Fit value function $\psi_{w+1} = \arg\min_\psi \frac{1}{|\mathcal{D}_w|} \sum_{G,\boldsymbol{a}} \left(\text{Value}_\psi(G) - \mathcal{R}^w\right)^2$.
16: **end for**

---

allowing arbitrary locations, thereby reflecting practical deployment considerations in real-world scenarios. The objective is formally defined as

$$\text{Maximize } \mathbb{E}_{\boldsymbol{a} \sim \pi_\theta}[\mathcal{R}(\boldsymbol{a} \mid G)] \quad \text{subject to} \quad \sum_{i=1}^{|V|} \boldsymbol{a}_i = k\,|V|. \tag{3}$$

We utilize the Proximal Policy Optimization (PPO) algorithm (Schulman et al., 2017), and decompose this problem into two parts: (1) How to sample from $\pi_\phi$ while respecting the constraints and (2) How to compute the log probability $\log \pi_\phi(\mathbf{a} \mid \sum_{i=1}^{|V|} \boldsymbol{a}_i = k|V|)$.

### 4.1 Constrained Sampling

We first address the problem of sampling from the constrained distribution. Sampling from discrete distributions has been investigated in the literature. In particular, the Gumbel-softmax approach (Jang et al., 2017; Maddison et al., 2017) leverages continuous relaxations to reparameterize the categorical distribution by perturbing the class logits with Gumbel noise and passing them through a temperature-scaled softmax. Extending this idea, reparameterizable subset sampling (Xie & Ermon, 2019) generalizes the Gumbel-softmax trick to $k$-subset sampling, thereby rendering it amenable to backpropagation. Moreover, recent work by Ahmed et al. (2023) employs dynamic programming to sample exactly from the constrained distribution.

The sampling strategy from Ahmed et al. (2023) necessitates constructing a dynamic programming table of $\pi_\phi(\sum_{i=1}^{|V|} \boldsymbol{a}_i = k|V|)$ with varying $|V|$ and $k|V|$, which runs in $\mathcal{O}(k|V|^2)$ complexity. Then, the sampling algorithm runs in $\mathcal{O}(|V|)$ complexity. We refer the readers to Ahmed et al. (2023) for additional details. However, this method becomes computationally infeasible when applied to meshes with many nodes. Consequently, we adopt the Gumbel approach with top-$k|V|$ selection (Kool et al., 2020). For each node $v_i$, we sample independent Gumbel noise and compute the perturbed scores:

$$s_i = \log \boldsymbol{p}_i + g_i \quad \text{with } g_i \sim \text{Gumble}(0, 1). \tag{4}$$

Then, we select the indices corresponding to the top-$k|V|$ largest values:

$$\boldsymbol{a} = \text{Top-}k|V| \text{ indices of } \{s_i\}_{i=1}^{|V|}. \tag{5}$$

This procedure runs in $\mathcal{O}(|V| \log k|V|)$ with min-heap streaming. Since PPO does not require differentiation through the sample, we avoid the $\mathcal{O}(k|V|^2)$ complexity incurred by differentiable Gumble $k|V|$-subset sampling by Xie & Ermon (2019).

Assuming perfect parallelization, the vectorized complexity of computing the dynamic programming table in Ahmed et al. (2023) can reach $\mathcal{O}(\log k|V| \log |V|)$, while sampling achieves $\mathcal{O}(\log |V|)$. In

contrast, the Gumbel approach with top-$k|V|$ selection attains $\mathcal{O}(\log |V|)$ vectorized complexity, offering a significantly faster alternative.

## 4.2 Constrained Log Probability

We now address the problem of computing the log probability $\log \pi_\phi(\boldsymbol{a} \mid \sum_{i=1}^{|V|} \boldsymbol{a}_i = k|V|)$. By Bayes' rule, it can be expressed as $\log \pi_\phi(\boldsymbol{a} \mid \sum_{i=1}^{|V|} \boldsymbol{a}_i = k|V|) = \log \pi_\phi(\boldsymbol{a})[\sum_{i=1}^{|V|} \boldsymbol{a}_i = k|V|] - \log \pi_\phi(\sum_{i=1}^{|V|} \boldsymbol{a}_i = k|V|)$, where $[\sum_{i=1}^{|V|} \boldsymbol{a}_i = k|V|]$ denotes an indicator function. The term $\log \pi_\phi(\sum_{i=1}^{|V|} \boldsymbol{a}_i = k|V|)$ appears intractable (Ahmed et al., 2023), since a brute force computation would entail $\mathcal{O}(\binom{|V|}{k|V|})$ complexity. An efficient method proposed by Ahmed et al. (2023) leverages dynamic programming to compute this log probability exactly with $\mathcal{O}(k|V|^2)$. However, even this improvement remains computationally prohibitive within the context of our problem.

To address this issue, we employ the saddle point approximation (Daniels, 1954) to estimate the log probability $\log \pi_\phi(\sum_{i=1}^{|V|} \boldsymbol{a}_i = k|V|)$. The saddle point approximation provides a highly accurate method for approximating any probability distribution function and is particularly effective for the distribution of the sum of independent random variables. In the Bernoulli setting, where there are $|V|$ independent Bernoulli variables with parameters $\boldsymbol{p}_i$, we define the cumulant generating function as

$$\psi(t) = \sum_{i=1}^{|V|} \log \left(1 - \boldsymbol{p}_i + \boldsymbol{p}_i e^t\right). \tag{6}$$

Consequently, the probability can be approximated by

$$\pi_\phi(\sum_{i=1}^{|V|} \boldsymbol{a}_i = k|V|) \approx \frac{1}{\sqrt{2\pi \, \psi''(t^*)}} \exp\left(\psi(t^*) - k \, t^*\right), \tag{7}$$

where $t^*$ is the saddle point found by solving $\psi(t^*) = k|V|$. In practice, it is determined using a differentiable numerical root-finding algorithm, such as Newton-Raphson, over a finite number of iterations. We present the approximation complexity below. The linear complexity and vectorized log complexity are very favorable in our problem setting, where the number of mesh points could be extremely large. We refer the readers to Appendix D for detailed proof.

**Proposition 1** (Saddle Point Approximation Complexity). *Suppose $\boldsymbol{a}_i \sim Bernoulli(\boldsymbol{p}_i)$. When the probability $\pi_\phi(\sum_{i=1}^{|V|} \boldsymbol{a}_i = k|V|)$ is approximated using the saddle point method, the algorithmic complexity of computing $\pi_\phi(\boldsymbol{a} \mid \sum_{i=1}^{|V|} \boldsymbol{a}_i = k|V|)$ is $\mathcal{O}(|V|)$. Assuming perfect parallelization, the vectorized complexity can achieve $\mathcal{O}(\log |V|)$*

Additionally, we establish an error bound for the approximation, with the proof presented in Appendix E. This bound indicates that the relative error diminishes as the number of nodes grows, which is especially beneficial for our problem.

**Proposition 2** (Saddle Point Approximation Error Bound). *Let $\boldsymbol{a}_i \sim Bernoulli(\boldsymbol{p}_i)$. Then, the asymptotic relative error in $\pi_\phi(\boldsymbol{a} \mid \sum_{i=1}^{|V|} \boldsymbol{a}_i = k|V|)$ when approximating $\pi_\phi(\sum_{i=1}^{|V|} \boldsymbol{a}_i = k|V|)$ using the saddle point approximation is given by $\pi_\phi(\boldsymbol{a} \mid \sum_{i=1}^{|V|} \boldsymbol{a}_i = k|V|) = \hat{\pi}_\phi(\boldsymbol{a} \mid \sum_{i=1}^{|V|} \boldsymbol{a}_i = k|V|) \left(1 - O\left(\frac{1}{|V|}\right)\right)$, where $\pi_\phi(\boldsymbol{a} \mid \sum_{i=1}^{|V|} \boldsymbol{a}_i = k|V|)$ denotes the ground truth probability and $\hat{\pi}_\phi(\boldsymbol{a} \mid \sum_{i=1}^{|V|} \boldsymbol{a}_i = k|V|)$ denotes our approximated probability.*

## 4.3 Algorithm

With (1) sampling from the constrained distribution and (2) estimating the log probability addressed, the intuitive solution is to integrate them into the standard PPO algorithm. However, at such a high dimensionality, we observed that initiating the training with a constrained policy for a randomly initialized model is overly restrictive, leading to a failure to learn. Consequently, we propose a *Two-Step Constrained PPO* training procedure.

In the first stage, the PPO is trained in an unconstrained manner, meaning that both the sampling and log probability computations are performed without enforcing any constraints. To softly enforce the constraints, we adopt a penalized objective function as presented in Proposition 3.

**Proposition 3** (Penalized Reward Function (from Liu et al. (2024))). *Let $\boldsymbol{a} \in \{0, 1\}^n$ and assume that constraint is $\mathbf{1}^T \boldsymbol{a} = k|V|$. Assume the reward function $\mathcal{R}$ is Lipschitz with respect to $\boldsymbol{a}$. Then,*

*if $\boldsymbol{a}^*$ optimizes the reward $\mathcal{R}$, it also optimizes $\hat{\mathcal{R}} = \mathcal{R} - \lambda(k|V| - ||\boldsymbol{a}||_2^2)^2$, for all $\lambda > 0$, and the optimal reward of $\mathcal{R}$ is equivalent to $\hat{\mathcal{R}}$.*

Although converting a constrained optimization problem into its penalized form has been widely studied (Boyd & Vandenberghe, 2004; Bertsekas, 1999), these approaches typically require the convexity of the original objective function to ensure equivalence in the global optimizer. In light of recent work on non-convex optimization theory (Liu et al., 2024), we show that as long as the objective function, in this case, $\mathcal{R}$, is Lipschitz, the penalization formulation ensures the global maximizer of $\hat{\mathcal{R}}$ is also the global maximizer for $\mathcal{R}$, which provides a theoretical guarantee for our training procedure. Since $\mathcal{R}$ represents MSE of the reconstructed data from our reconstruction network $\mathcal{M}_\theta$, the Lipschitz assumption merely requires that the gradient of the MSE is differentiable with respect to the input of $\mathcal{M}_\theta$. This is a reasonable assumption given that training $\mathcal{M}_\theta$ demands non-exploding gradients, and the Lipschitz condition holds in our experimental domain.

After training under the penalized reward setting for $T_1$ iterations, we switch to constrained training, where we sample $\boldsymbol{a}$ from the constrained distribution using the Gumble Top-$k|V|$ discussed in Section 4.1 and compute the constrained log probability using saddle point approximation in Section 4.2. The complete algorithm is provided in Algorithm 1. During inference, we sample from the constrained distribution using Gumble Top-$k|V|$.

## 5 Experiment

**Dataset** Four three-dimensional turbulent flow datasets were generated using DNS in COMSOL (COM, 2020). The datasets comprise variations in four distinct geometries: (1) *Sphere* with variable radius, (2) *Ellipsoid* with varying semi-axis lengths, (3) *Cylinder* with varying height and radius, and (4) *NACA 4-digit* airfoil with varying chord length and thickness. The initial velocity field was randomly generated, while the inlet velocity was modeled as time-dependent and stochastic, expressed as $\boldsymbol{u}(\boldsymbol{x}, t) = f(\boldsymbol{x}, t, \Theta) + h(\boldsymbol{x}, \Theta)\epsilon_t$, where $f(\boldsymbol{x}, t, \Theta)$ specifies the prescribed inlet velocity at each spatial location and $h(\boldsymbol{x}, \Theta)$ controls the magnitude of the random noise term. $\Theta$ contains the parameters of the geometries. Although these datasets were generated by numerical simulation, they effectively mimic the dynamic behavior observed in actual turbulent flows. Code and datasets are available at Github.

**Task Setup and Baselines** We evaluate our proposed directional transport-aware GNN (DTA-GNN) on all datasets under varying sensor placement configurations. Specifically, two sensor distribution strategies are considered: (i) Uniform, in which sensors are evenly distributed across the computational domain, and (ii) Random, in which sensor locations are selected randomly. For each strategy, sensor densities of 5%, 10%, 20%, and 30% of the total mesh points are used.

To benchmark our approach, we compare it against several baselines. First, two naive interpolation methods, Mean and k-Nearest Neighbors (KNN), are included. Additionally, we compare with DiffusionPDE (Huang et al., 2024a) and OFormer (Li et al., 2023), both of which have demonstrated strong performance in PDE forward modeling with partial-observation. Considering that our data are mesh-based, we also include MeshGraphNets (Pfaff et al., 2021) and Graph Kernel Operator (GKO) (Li et al., 2020), which are recognized for their excellent performance in mesh-based PDE forward simulation. Moreover, Flow Completion Network (FCN) (He et al., 2022), designed specifically for sparse sensor measurement interpolation, is also considered in our experiments. The details of these baseline implementations are provided in the Appendix C. Performance is quantified using the MSE of the normalized velocity and pressure fields $\text{MSE}([\hat{\boldsymbol{u}}_{\text{normalized}}, \hat{\boldsymbol{p}}_{\text{normalized}}], [\boldsymbol{u}_{\text{normalized}}, \boldsymbol{p}_{\text{normalized}}])$.

### 5.1 Main Results

We present the results in Table 2. Across all four geometries and under both uniform and random sensor placement strategies, our model consistently achieves the lowest reconstruction error, often by a wide margin. We also observe that several baselines, including FCN, DiffusionPDE, and MeshGraphNets, achieve strong performance under particular sensor placement strategies on certain datasets. However, our reconstruction model is able to achieve consistent improvements across different shapes, sensor densities, and distributions, which demonstrates that our approach more effectively captures underlying flow field and yields robust predictions even in highly under-sampled regimes. In the most challenging *NACA 4-digit* scenario, our model is able to achieve approximately 10% improvements in many placement strategies.

| | | Mean | KNN | OFormer | GKO | FCN | MeshGraphNets | DiffusionPDE | DTA-GNN |
|---|---|---|---|---|---|---|---|---|---|
| *Sphere* | 5% Uniform | $\geq 10^4$ | 221.413 | 9.571 | 10.819 | 8.367 | 5.612 | 9.923 | **5.534** |
| | 10% Uniform | $\geq 10^4$ | 121.119 | 4.400 | 6.745 | 6.153 | 3.537 | 3.755 | **3.092** |
| | 20% Uniform | $\geq 10^4$ | 72.916 | 7.285 | 4.877 | 4.856 | 4.280 | 2.283 | **1.724** |
| | 30% Uniform | $\geq 10^4$ | 53.369 | 2.532 | 4.346 | 4.754 | 3.537 | 1.775 | **1.204** |
| | 5% Random | $\geq 10^4$ | 419.095 | 18.614 | 15.879 | 9.654 | 8.741 | 8.978 | **7.180** |
| | 10% Random | $\geq 10^4$ | 220.894 | 10.494 | 12.410 | 7.655 | 4.541 | 5.872 | **4.110** |
| | 20% Random | $\geq 10^4$ | 120.537 | 6.460 | 7.305 | 6.998 | 2.778 | 2.730 | **2.155** |
| | 30% Random | $\geq 10^4$ | 88.131 | 3.681 | 5.121 | 4.927 | 1.563 | 1.687 | **1.446** |
| *Ellipsoid* | 5% Uniform | $\geq 10^4$ | 248.818 | 28.395 | 63.162 | 69.162 | 53.003 | 68.809 | **21.306** |
| | 10% Uniform | $\geq 10^4$ | 136.951 | 20.380 | 49.931 | 41.602 | 18.105 | 43.315 | **9.676** |
| | 20% Uniform | $\geq 10^4$ | 85.819 | 17.231 | 45.390 | 28.203 | 9.612 | 14.055 | **7.083** |
| | 30% Uniform | $\geq 10^4$ | 67.111 | 16.384 | 42.654 | 15.146 | 9.112 | 12.128 | **6.310** |
| | 5% Random | $\geq 10^4$ | 492.094 | 60.837 | 140.946 | 41.612 | 53.010 | 99.697 | **39.033** |
| | 10% Random | $\geq 10^4$ | 238.163 | 25.683 | 73.284 | 29.975 | 27.494 | 57.804 | **14.175** |
| | 20% Random | $\geq 10^4$ | 132.324 | 16.882 | 53.019 | 10.513 | 15.684 | 11.450 | **10.116** |
| | 30% Random | $\geq 10^4$ | 98.768 | 15.197 | 48.623 | 10.267 | 10.705 | 7.728 | **5.930** |
| *Cylinder* | 5% Uniform | $\geq 10^4$ | 195.235 | 24.665 | 32.694 | 25.983 | 16.684 | 24.340 | **15.901** |
| | 10% Uniform | $\geq 10^4$ | 124.619 | 21.659 | 27.479 | 21.511 | 8.353 | 16.612 | **7.932** |
| | 20% Uniform | $\geq 10^4$ | 53.977 | 14.011 | 24.583 | 16.765 | 2.622 | 5.310 | **2.299** |
| | 30% Uniform | $\geq 10^4$ | 28.973 | 6.368 | 23.610 | 11.656 | 2.331 | 3.311 | **1.913** |
| | 5% Random | $\geq 10^4$ | 355.989 | 51.832 | 48.321 | 53.574 | 46.980 | 43.895 | **43.723** |
| | 10% Random | $\geq 10^4$ | 195.992 | 15.717 | 21.446 | 28.640 | 14.091 | 14.193 | **12.797** |
| | 20% Random | $\geq 10^4$ | 108.420 | 8.732 | 12.358 | 26.532 | 6.123 | 6.346 | **5.765** |
| | 30% Random | $\geq 10^4$ | 73.905 | 7.091 | 10.955 | 18.663 | 2.895 | 3.134 | **2.152** |
| *NACA 4-digit* | 5% Uniform | $\geq 10^4$ | 1735.205 | 65.524 | 87.756 | 102.914 | 78.421 | 72.974 | **59.308** |
| | 10% Uniform | $\geq 10^4$ | 1110.690 | 44.384 | 72.311 | 88.425 | 74.549 | 66.277 | **43.647** |
| | 20% Uniform | $\geq 10^4$ | 769.226 | 31.714 | 54.879 | 52.877 | 32.596 | 50.627 | **28.173** |
| | 30% Uniform | $\geq 10^4$ | 656.289 | 30.634 | 46.273 | 49.116 | 29.817 | 44.358 | **24.883** |
| | 5% Random | $\geq 10^4$ | 2623.504 | 148.324 | 156.791 | 113.718 | 103.126 | 97.781 | **97.076** |
| | 10% Random | $\geq 10^4$ | 1438.934 | 57.024 | 92.245 | 71.875 | 62.337 | 60.306 | **56.224** |
| | 20% Random | $\geq 10^4$ | 1028.617 | 46.368 | 77.994 | 55.630 | 48.519 | 49.497 | **44.748** |
| | 30% Random | $\geq 10^4$ | 816.640 | 29.923 | 55.475 | 44.426 | 31.806 | 29.912 | **29.803** |

Table 2: Comparison MSE scaled by $10^{-4}$ across multiple datasets and sensor placement schemes.

**Parameter Efficiency** Besides the performance gain, we also show that our model is parameter efficient. We present the number of parameters for various models in Table 3.

## 5.2   Sea Surface Temperature

We evaluate our reconstruction model on NOAA OISST V2 weekly mean sea surface temperature dataset recorded from December 31, 1989 through January 29, 2023 (Reynolds et al., 2008) at a $1° \times 1°$ spatial resolution and with sensors at $10\%$ of the grid locations. We train on $80\%$ of the data, use $10\%$ for validation and the remaining $10\%$ for testing. We compare with three strong baselines from Section 5.1: FCN, DiffusionPDE, and MeshGraphNets. As shown in Table 2, our approach achieves a relative improvement of $14.01\%$ in MSE over the strongest competing method DiffusionPDE.

| Model | MSE |
|---|---|
| FCN | 15.761 |
| DiffusionPDE | 7.969 |
| MeshGraphNets | 8.176 |
| DTA-GNN | **6.853** |

Figure 2: Comparison of different models on the global ocean surface temperature dataset. All numbers are scaled by $10^{-2}$.

Figure 5 illustrates representative reconstructions for both summer and winter seasons. In addition to accurately recovering the large-scale seasonal cycle and major ocean gyre structures, our model

also succeeds at resolving the much smaller-scale temperature anomalies that arise from equatorial upwelling, coastal current meanders, and tropical instability waves. These fine-scale features play an outsized role in modulating air–sea heat fluxes and driving interannual phenomena such as El Niño–Southern Oscillation. Our model accurately reconstructs both the global trend and these small, dynamically driven variations with high fidelity. We present additional visualizations in Appendix N.

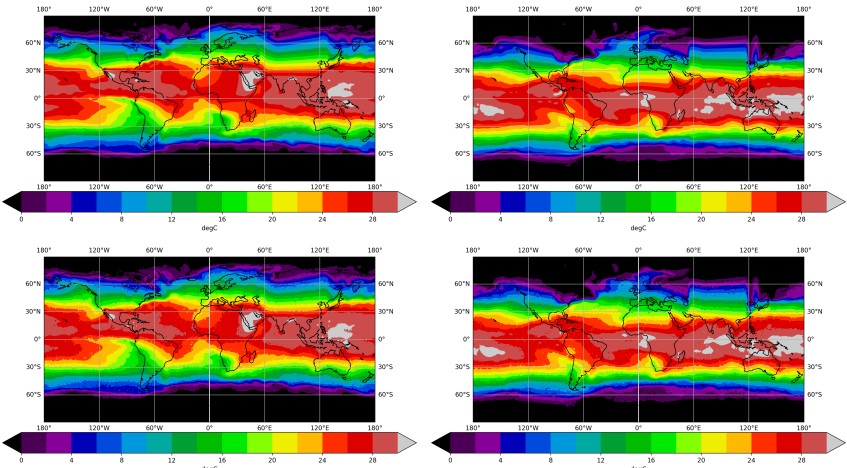

Figure 3: Visualization of ground truth sea surface temperature data and reconstructed sea surface temperature data. The first row corresponds to ground truth data and the second row corresponds to reconstructed data from DTA-GNN.

## 5.3 Optimal Sensor Placement

We further enhance the reconstruction accuracy of our model by identifying optimal sensor locations. After training the reconstruction model $\mathcal{M}_\theta$ as described in Section 5.1, we incorporate it into the reward function of the *Two Step Constrained PPO*. We refer the readers to Appendix F for training and model hyperparameters. We consider two standard baselines, uniform placement and random placement, and assume that $10\%$ of the mesh points have sensors.

Figure 4 reports the MSE of the reconstructed data for each method. Our sensor placement policy achieves approximately a $15\%$ reduction in MSE relative to uniformly placed sensors. The performance gain can be attributed to concentrating sensors in regions of fluid high variability, such as high velocity and pressure gradient. Additionally, our policy accounts for performance disparity in reconstruction models. By explicitly optimizing for both criteria, our policy achieves significant improvements in reconstruction accuracy.

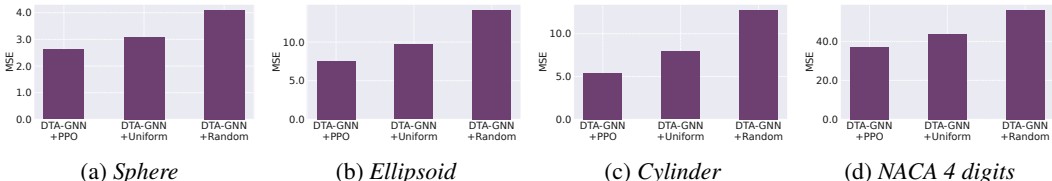

| (a) *Sphere* | (b) *Ellipsoid* | (c) *Cylinder* | (d) *NACA 4 digits* |

Figure 4: Comparison of sensor placement schemes across datasets. We assume that $10\%$ of the mesh points have sensors. All numbers are scaled by $10^{-4}$.

## 6 Conclusion

We introduce a realistic problem formulation for fluid-field reconstruction and design a directional transport–aware GNN that achieves superior reconstruction accuracy across multiple datasets and sensor-placement configurations. We observe that conventional sensor placement algorithms often fail to identify optimal sensor locations, and we propose a *Two Stage Constrained PPO* training procedure to train a sensor placement policy that yields additional improvements.

# 7 Acknowledgments

This work was partially supported by NSF Center for Computer Assisted Synthesis (2202693), National Artificial Intelligence Research Resource (NAIRR) Pilot (240280, 240443), National Science Foundation (2106859, 2211557, 2119643, 2200274, 2303037, 2312501, 2531008), National Institutes of Health (U54HG012517, U24DK097771, U54OD036472), NEC, Optum AI, SRC JUMP 2.0 Center, Amazon Research Awards, and Snapchat Gifts.

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

# A    Limitations

Our work uses datasets simulated with DNS in Comsol and adopts time-dependent and stochastic inlet velocity to better reflect real-world scenarios. Since there are no large-scale CFD datasets collected from the real world, we believe that our simulated datasets better represent real-world scenarios. However, we acknowledge that models trained using such simulated datasets could have performance degradation when inferring on real-world datasets. The performance degradation could be attributed to the inaccuracies from real-world sensors. Future work could focus on conducting extensive real-world experiments to collect real-world datasets and enhance the real-world applicability of related works.

# B    Broader Impacts

Our reconstruction framework and optimal sensor placement strategy offer transformative potential across aerospace, automotive, and environmental engineering. In the aerospace sector, for example, it can dramatically reduce wind tunnel testing time by inferring complete flow fields from just a handful of strategically positioned probes. In automotive design, it paves the way for rapid, cost-effective aerodynamic optimization by filling in the gaps between sparse on-vehicle measurements.

By accurately reconstructing full CFD fields from partial observations, our approach lowers both the logistical and financial barriers to advanced flow analysis. It enables engineers and researchers to iterate designs more quickly, run fewer physical experiments, and integrate real-time monitoring into digital-twin platforms. Ultimately, this work democratizes access to high-fidelity flow data, accelerates research and development cycles, and fosters deeper scientific insight across multiple disciplines.

# C    Baseline and Model Implementation Detail

**OFormer** We adopt the model from Li et al. (2023). The hidden dimension is set to 64. We add a separate MLP in the encoder to encode the binary mask indicating sensor placement. The mask encoder latent dimension is set to 16. The number of attention blocks is set to 2. The models are trained with batch size 16 for 300 epochs. We use a cosine annealing learning rate with a starting learning rate at $1 \times 10^{-4}$ to an end learning rate at $1 \times 10^{-5}$.

**GKO** We adopt the model from Li et al. (2020). We add a separate MLP in the encoder to encode the binary mask indicating sensor placement. The mask encoder latent dimension is set to 16. We choose width=256 and depth=6. The models are trained with batch size 16 for 300 epochs. We use a cosine annealing learning rate with a starting learning rate of $1 \times 10^{-4}$ to end learning rate at $1 \times 10^{-5}$.

**FCN** We adopt the model from He et al. (2022). The model consists of three graph convolution layers and two spatial gradient attention layers. The latent dimension is set to 64 with a separate MLP to encode the binary mask indicating sensor placement. The mask encoder latent dimension is set to 16. The models are trained with batch size 16 for 300 epochs. We use a cosine annealing learning rate with a starting learning rate at $1 \times 10^{-4}$ to an end learning rate at $1 \times 10^{-5}$.

**DiffusionPDE** We adopt the model from Huang et al. (2024a). The score network in DiffusionPDE has been modified by incorporating message-passing graph neural networks, addressing the limitation that the original UNet architecture in DiffusionPDE is not directly compatible with mesh data. We include 6 message passing layers with latent size 64. The encoder is the same as our methods. We use 4 layers of MLPs with relu activation functions for each encoder block. The latent dimension for the encoding of the binary mask is 16 and the other encoder's latent dimensions are 64. For diffusion parameters, $\beta_{\min} = 0.0001$ and $\beta_{\max} = 0.02$. The total number of diffusion steps is 1000. The models are trained with batch size 16 for 300 epochs. We use a cosine annealing learning rate with a starting learning rate at $1 \times 10^{-4}$ to an end learning rate at $1 \times 10^{-5}$.

**MeshGraphNets** We adopt the model from Pfaff et al. (2021). We include 6 message passing layers with latent size 64. The encoder is the same as our methods. We use 4 layers of MLPs with relu activation functions for each encoder block. The latent dimension for the encoding of the binary mask is 16 and the other encoder's latent dimensions are 64. The decoder consists of a single 4-layer MLP with relu activation functions. The models are trained with batch size 16 for 300 epochs. We

use a cosine annealing learning rate with a starting learning rate of $1 \times 10^{-4}$ to end learning rate at $1 \times 10^{-5}$.

**Ours** We include 6 message passing layers with latent size 64. We use 4 layers of MLPs with relu activation functions for each encoder block. The latent dimension for the encoding of the binary mask is 16 and the other encoder's latent dimensions are 64. The decoder consists of a single 4-layer MLP with relu activation functions. The models are trained with batch size 16 for 300 epochs. We use a cosine annealing learning rate with a starting learning rate at $1 \times 10^{-4}$ to an end learning rate at $1 \times 10^{-5}$.

**Model Parameters** We present the number of parameters for different models in the following table:

| Model | # parameters |
|---|---|
| OFormer | 727,316 |
| GKO | 336,228 |
| FCN | 388,836 |
| DiffusionPDE | 317,236 |
| MeshGraphNets | 315,412 |
| Ours | **266,260** |

Table 3: Comparison of parameter counts for various models.

# D    Proof of Proposition 1

Suppose $\boldsymbol{a}_i \sim \text{Bernoulli}(\boldsymbol{p}_i)$. The cumulative generating function and its derivatives have closed form:

$$\psi(t) = \sum_{i=1}^{|V|} \log\left(1 - \boldsymbol{p}_i + \boldsymbol{p}_i e^t\right) \tag{8}$$

$$\psi'(t) = \sum_{i=1}^{|V|} \frac{\boldsymbol{p}_i e^t}{1 - \boldsymbol{p}_i + \boldsymbol{p}_i e^t} \tag{9}$$

$$\psi''(t) = \sum_{i=1}^{|V|} \frac{\boldsymbol{p}_i e^t (1 - \boldsymbol{p}_i)}{(1 - \boldsymbol{p}_i + \boldsymbol{p}_i e^t)^2} \tag{10}$$

$$\tag{11}$$

Their evaluations all cost $\mathcal{O}(|V|)$. Newton's method converges quadratically, so to reach an error tolerance $\epsilon$, we need $\mathcal{O}(\log\log(\frac{1}{\epsilon}))$. Thus, the overall runtime is $\mathcal{O}(|V|\log\log(\frac{1}{\epsilon}))$. Since we specify a fixed precision and a maximum number of iterations (this is chosen to be a small number, since Newton Raphson generally converges fast), the runtime becomes $\mathcal{O}(|V|)$. For vectorized computation, computing $\psi(t)$, $\psi'(t)$, and $\psi''(t)$ take $\mathcal{O}(\log|V|)$ complexity and the overall runtime becomes $\mathcal{O}(\log|V|)$.

# E    Proof of Proposition 2

Let $\boldsymbol{a}_i \sim \text{Bernoulli}(\boldsymbol{p}_i)$. Then,

$$p(\boldsymbol{a} \mid \sum_{i=1}^{|V|} \boldsymbol{a}_i = k|V|) = \frac{p(\boldsymbol{a})[\sum_{i=1}^{|V|} \boldsymbol{a}_i = k|V|]}{p(\sum_{i=1}^{|V|} \boldsymbol{a}_i = k|V|)}$$

In the standard asymptotic regime, the relative error in the likelihood estimated using the saddle point approximation is of order $\frac{1}{|V|}$ (Goodman, 2022). Then,

$$p(\boldsymbol{a} \mid \sum_{i=1}^{|V|} \boldsymbol{a}_i = k|V|) \approx \frac{p(\boldsymbol{a})[\sum_{i=1}^{|V|} \boldsymbol{a}_i = k|V|]}{p(\sum_{i=1}^{|V|} \boldsymbol{a}_i = k|V|)(1 + \mathcal{O}(\frac{1}{|V|}))}$$

$$\approx p(\boldsymbol{a} \mid \sum_{i=1}^{|V|} \boldsymbol{a}_i = k|V|)\frac{1}{1 + \mathcal{O}(\frac{1}{|V|})}$$

By Taylor Expansion,

$$\hat{p}(\boldsymbol{a} \mid \sum_{i=1}^{|V|} \boldsymbol{a}_i = k|V|) = p(\boldsymbol{a} \mid \sum_{i=1}^{|V|} \boldsymbol{a}_i = k|V|)(1 - \mathcal{O}(\frac{1}{|V|}))$$

## F   Hyperparameters for Optimal Sensor Placement

The actor and critic network consists of 6 layers of message-passing GNNs with latent dimension 128. The value predicted by the critic is taken as the mean of the decoded graph. $\lambda$ from the penalized reward function is taken as 0.00015. The number of gradient descent steps is 5, and we adopt a clip value of 0.2. We train under the penalized scheme for 1500000 steps ($T_1 = 1500000$) and under the constrained scheme for 500000 steps ($T_2 = 2000000$). We employ a cosine learning rate for both actor and critic with a starting learning rate $1 \times 10^{-5}$ and end learning rate $1 \times 10^{-7}$.

## G   Additional Experiment Results on Optimal Sensor Placement

Techniques that determine sensor placement through QR pivoting and SVD assume that high-dimensional states can be effectively represented by latent low-dimensional structures, an inherent compressibility that enables sparse sensing. In particular, a high-dimensional state $\boldsymbol{x} \in \mathbb{R}^n$ is often assumed to have a compact representation in a suitable transform basis $\boldsymbol{\Phi} \in \mathbb{R}^{n \times r}$, so that $\boldsymbol{x} = \boldsymbol{\Phi s}$, where $\boldsymbol{s} \in \mathbb{R}^r$ is sparse and and $r < n$. The objective is to design a measurement matrix $\boldsymbol{C} \in \mathbb{R}^{p \times n}$, with a small number $p \leq n$ of optimized measurements, such that the measurement vector $\boldsymbol{y} = \boldsymbol{Cx} \in \mathbb{R}^p$ enables accurate reconstruction of $\boldsymbol{s}$, and consequently $\boldsymbol{x}$.

We provide the reconstructed MSE in Table 4 tested in the *Sphere* dataset. We notice that the reconstruction MSE from sensor locations determined by QR Pivoting and d-optimal is significantly higher than uniform or randomly placed sensors. Note that for randomly placed sensors, we randomly sample sensor locations for every frame of the data and then feed it into the reconstruction network, whereas the sensor locations for QR Pivoting and d-optimal are fixed for the entire trajectory.

| Method | MSE |
|---|---|
| Uniform | 3.092 |
| Random | 4.110 |
| QR Pivoting | 6.988 |
| d-optimal | 6.309 |

Table 4: Comparison of sensor locations on *Sphere* datasets with sensor density of $10\%$. We report the MSE scaled by $10^{-4}$.

## H   Independence Assumption of Random Variables

We assume that the random variables are independent. This assumption is reasonable because each sensor provides measurements only at its specific location, and the removal of one sensor does not directly influence the others. However, when the total number of sensors is constrained, statistical correlation is introduced through the equality constraint.

# I  Experiments on Turbulence Data

We compare our proposed DTA-GNN against three strong baselines, FlowCompletionNetwork (FCN), DiffusionPDE, and MeshGraphNets, on two benchmark datasets: Kolmogorov Flow and Taylor-Green Vortex. Both datasets are generated via high-resolution numerical simulations using a pseudo-spectral solver governed by the incompressible Navier–Stokes equations. The Kolmogorov Flow dataset features a time-dependent sinusoidal external forcing and is simulated at a Reynolds number of 2000. The Taylor-Green Vortex dataset is initialized from its analytical solution and perturbed with Gaussian noise to produce a variety of flow trajectories. Simulations are carried out at a Reynolds number of 1500. The performance of each method on these datasets is summarized in the table below:

Table 5: Performance comparison on Kolmogorov Flow.

| Method | 5% Random | 10% Random | 20% Random | 30% Random |
|---|---|---|---|---|
| Ours | 9.484 | 8.537 | 4.510 | 3.443 |
| FCN | 10.547 | 9.1557 | 5.109 | 4.690 |
| DiffusionPDE | 11.214 | 9.458 | 6.699 | 4.703 |
| MeshGraphNets | 10.180 | 9.283 | 5.271 | 4.137 |

Table 6: Performance comparison on Taylor Green Vortex.

| Method | 5% Random | 10% Random | 20% Random | 30% Random |
|---|---|---|---|---|
| Ours | 6.749 | 4.856 | 2.643 | 1.267 |
| FCN | 8.430 | 6.354 | 3.590 | 2.825 |
| DiffusionPDE | 8.898 | 5.348 | 3.142 | 1.974 |
| MeshGraphNets | 9.801 | 5.677 | 3.119 | 2.831 |

# J  Generalisability Experiments

We evaluate the generalization capability of our proposed DTA-GNN by training it on the Ellipsoid dataset and testing it on the Sphere dataset. Notably, the Ellipsoid dataset contains no sphere geometries, ensuring that the test domain represents a previously unseen configuration. Furthermore, the two datasets differ in their inlet velocity profiles, resulting in entirely distinct flow fields. Despite these differences, as shown in the table below, DTA-GNN demonstrates strong generalization performance under these shifted conditions. In all test scenarios, except for the 30% Random setting (x% Random refers to sensors randomly distributed at x% of the mesh points), DTA-GNN consistently outperforms all baseline models, maintaining superior accuracy on the unseen flow distributions.

| Dataset | MSE (trained on this dataset) | Generalization MSE | % Drop |
|---|---|---|---|
| 5% Random | 7.180 | 7.650 | 6.54% |
| 10% Random | 4.110 | 4.468 | 8.71% |
| 20% Random | 2.155 | 2.328 | 8.03% |
| 30% Random | 1.446 | 1.574 | 8.86% |

# K  Ablation Studies on DTA-GNN

We conduct ablation studies in the following table. ABL_d means DTA-GNN without directional information in the message passing stage. ABL_diff means DTA-GNN without the difference between neighboring latent states. We simply concatenate the neighboring latent states and multiply with the direction information. MeshGraphNets corresponds to removing both the directional information and difference between neighboring latent states.

Based on these results, we draw three key conclusions: (1) Directional information plays a critical role when sensor density is low or when sensors are placed randomly, as some regions may lack sufficient sensor coverage. (2) Relying solely on the difference between neighboring latent states is suboptimal, as this approach does not capture edge attributes or directional cues essential for accurate information transfer. (3) Simply concatenating neighboring latent states leads to a moderate drop in performance,

| Dataset | MeshGraphNets | ABL_d | ABL_diff | DTA-GNN |
|---|---|---|---|---|
| 5% Uniform | 78.421 | 75.124 | 63.162 | 59.308 |
| 10% Uniform | 74.549 | 59.064 | 52.294 | 43.647 |
| 20% Uniform | 32.596 | 30.368 | 30.660 | 28.173 |
| 30% Uniform | 29.817 | 26.923 | 27.497 | 24.883 |
| 5% Random | 103.126 | 155.466 | 101.878 | 97.076 |
| 10% Random | 62.337 | 73.355 | 60.536 | 56.224 |
| 20% Random | 48.519 | 49.210 | 48.622 | 44.748 |
| 30% Random | 31.806 | 32.389 | 30.943 | 29.803 |

indicating that explicitly computing transported information is beneficial. Nevertheless, since message passing networks can still approximate difference operators, the performance degradation is less severe than when directional information is entirely removed.

## L Ablation Studies on Two-Step Constrained PPO

We report the ablation studies on Two-Step Constrained PPO in the following table. Penalized PPO refers to the first stage, where we use a penalized objective function to guide the model toward the constraints. During inference, we use Gumble Top-k to strictly enforce the equality constraints. Constrained PPO refers to the second stage, where we use Gumble Top-k for sampling and saddle point approximation for computing the log probability. Two-Step PPO without Saddlepoint Approx refers to the original Two-Step Constrained PPO, but removing saddle point approximation in the second stage. In the second stage, we use unconstrained log probability.

| Shape | Penalized PPO | Constrained PPO | without Saddlepoint Approx |
|---|---|---|---|
| Sphere | 3.270 | 19.280 | 3.178 |
| Ellipsoid | 11.002 | 31.647 | 10.629 |
| Cylinder | 9.490 | 55.014 | 9.575 |
| Airfoil | 46.431 | 327.966 | 44.103 |

From these results, we draw several important conclusions: (1) As discussed in the main paper, initiating constrained training from a randomly initialized policy is overly restrictive and substantially limits the model's performance. (2) Although the penalized training in the first stage helps guide the model toward a reasonable sensor placement strategy, it does not strictly enforce the equality constraint. Consequently, when combined with a constraint-compliant sampling strategy during inference, its performance degrades. (3) Accurately computing the log probability in the second-stage constrained training is essential. Without the saddle point approximation, as in the Two-Step PPO without Saddle Point Approximation, the model fails to outperform even the baseline of uniformly placed sensors.

## M Hardware Specification

We implement all models in PyTorch. All experiments are run on servers/workstations with the following configuration:

- 80 CPUs, 503G Mem, 8 x NVIDIA V100 GPUs.
- 48 CPUs, 220G Mem, 8 x NVIDIA TITAN Xp GPUs.
- 96 CPUs, 1.0T Mem, 8 x NVIDIA A100 GPUs.
- 64 CPUs, 1.0T Mem, 8 x NVIDIA RTX A6000 GPUs.
- 224 CPUs, 1.5T Mem, 8 x NVIDIA L40S GPUs.

# N Additional Experiment Results on Sea Surface Temperature

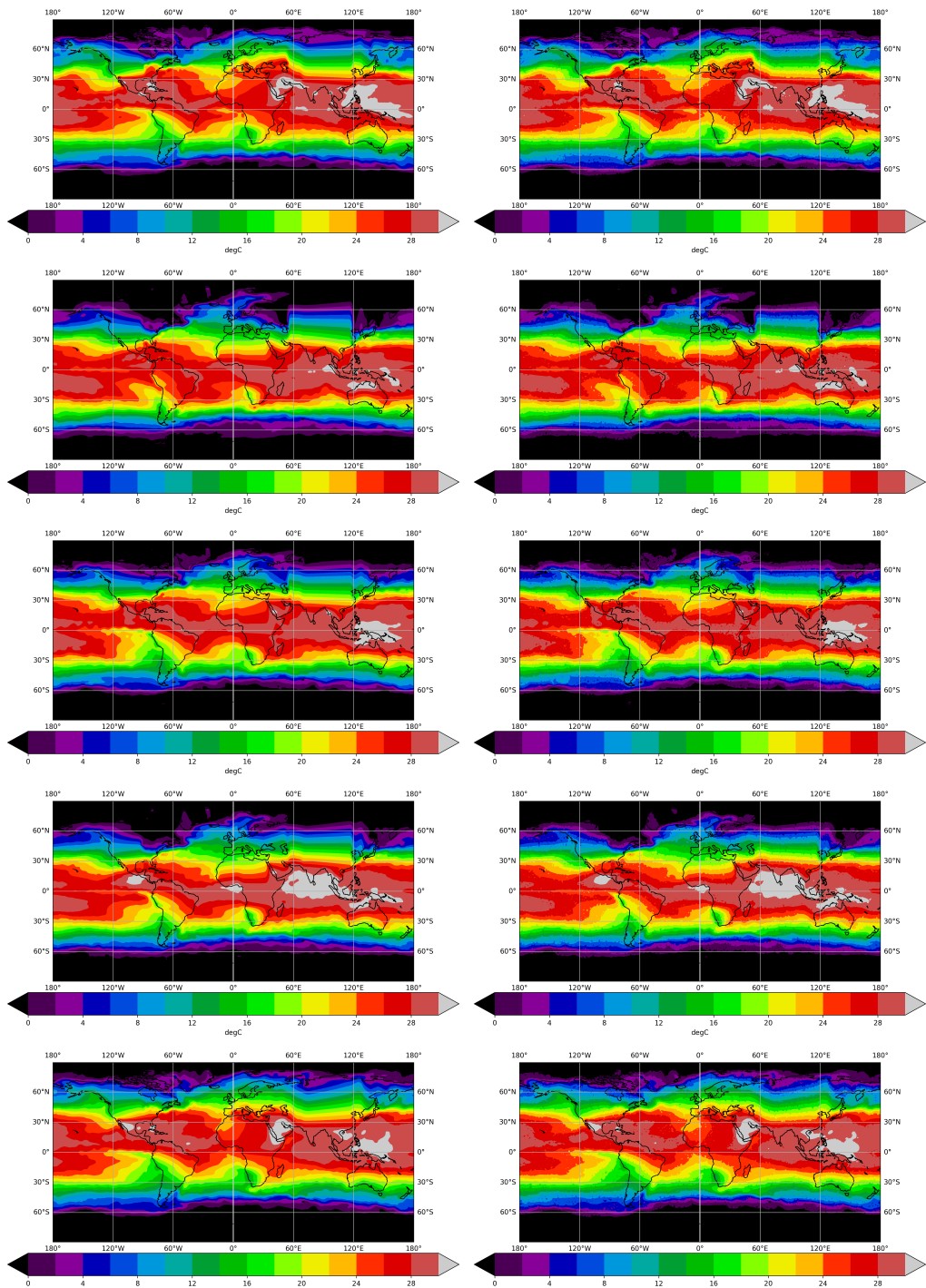

Figure 5: Visualization of ground truth sea surface temperature data and reconstructed sea surface temperature data. The first column corresponds to ground truth data and the second column corresponds to reconstructed data from our reconstruction model.

