# OpenReview forum: "Flow Field Reconstruction with Sensor Placement Policy Learning"
_NeurIPS.cc/2025/Conference — NeurIPS 2025 poster_

### Official Review · Reviewer_4m3m · 2025-06-03

**Clarity:** 3
**Significance:** 3
**Originality:** 3
**Rating:** 4
**Confidence:** 3

**Summary:**

Flow field reconstruction refers to the process of constructing complete flow field information from sparse sensor measurements. This is a crucial challenge in modern fluid dynamics because practical limits on sensor deployment often make it difficult to obtain high-fidelity, complete data. This paper develops a directional transport-aware Graph Neural Network (GNN) that explicitly encodes both directionality and information transport to address flow field reconstruction on arbitrary mesh-based geometries. Secondly, it finds that conventional sensor placement algorithms are insufficient in identifying effective sensor locations and thus proposes a novel Two-Step Constrained Proximal Policy Optimization (PPO) training strategy to learn a policy that identifies optimal sensor placement. Comprehensive experiments were conducted under realistic assumptions to validate the superiority of its reconstruction model and sensor placement policy, and it was tested on real-world meteorological data (sea surface temperature), outperforming existing methods.

**Questions:**

1. Scope of Real-World Validation: Address concerns about performance generalization based on the current extent of real-world dataset testing (e.g., primarily one dataset).
2. Ablation Studies: Clarify the impact of individual components in the proposed GNN (e.g., directional transport awareness) and the Two-Step Constrained PPO (e.g., the necessity of the two stages, saddlepoint approximation).
3. Comparison with Navier-Stokes Based Benchmarks: Explain the positioning of the work relative to methods tested on datasets derived from idealized physics or known governing equations like Navier-Stokes.
4. Robustness to Sensor Noise and Sim-to-Real Gap: Discuss the model's expected performance with noisy or imperfect real-world sensor data, and strategies to mitigate the sim-to-real gap.
5. Computational Cost and Scalability of PPO: Provide further details on the training time, resource requirements, and scalability of the Two-Step Constrained PPO for sensor placement, especially for larger systems.
6. Interpretability: Comment on the interpretability of the learned GNN representations and the sensor placement strategies derived from the PPO policy.
7. Sensor Placement Constraints: Clarify if sensor placement by the PPO is restricted (e.g., to boundaries only, as with the DNS dataset generation) or applicable to any predefined node in the flow field.

**Ethical Concerns:**

["NO or VERY MINOR ethics concerns only"]

**Final Justification:**

Thanks for the responses from the authors. Most of my concerns have been resolved. However, the computational cost remains unclear. Therefore, I keep my rating as Borderline accept.

**Limitations:**

1. Sensitivity to Real-World Sensor Imperfections: The model's robustness to real-world sensor noise, calibration errors, or other data imperfections is not extensively evaluated, as training primarily utilized high-fidelity simulated data or clean sea surface temperature data.
2. Computational Cost of PPO Training: Training the GNN-based Proximal Policy Optimization (PPO) policy for sensor placement can be computationally intensive and time-consuming, particularly for very large and complex flow fields, given the extensive training steps involved.
3. Fixed Sensor Proportion in PPO: The current PPO framework optimizes sensor placement for a predefined proportion or number of sensors, rather than determining the optimal number of sensors itself.
4. Interpretability of Learned Models: Gaining deep physical insights into the decision-making processes of the directional transport-aware GNN and the sensor placement policy learned by PPO remains challenging.

**Quality:**

3

**Strengths And Weaknesses:**

Strengths
1. General Model for Flow Field Reconstruction: The paper introduces a commendable general model for flow field reconstruction that distinguishes itself by not relying on many simplifying assumptions prevalent in existing methods. For instance, it doesn't require predefined governing equations (like Navier-Stokes) or restrict itself to two-dimensional domains. The core of this is a directional transport-aware Graph Neural Network (GNN) specifically designed to operate on arbitrary mesh-based geometries and infer flow fields without prior knowledge of the underlying Partial Differential Equations (PDEs).
2. Novel Two-Step Constrained PPO for Sensor Placement: A significant contribution is the novel Two-Step Constrained Proximal Policy Optimization (PPO) procedure developed for optimizing sensor placement. This approach addresses the observation that conventional sensor placement strategies can be suboptimal when integrated with deep learning reconstruction models. It also tackles the inherent difficulty of training reinforcement learning policies under strict constraints (e.g., an exact number of sensors) by first employing a penalized objective function before transitioning to a hard-constrained optimization phase.
3. State-of-the-Art Performance: The proposed methods, combining the GNN reconstruction model and the PPO-based sensor placement, demonstrate state-of-the-art (SOTA) performance. This is evidenced by consistently achieving the lowest reconstruction errors across four different synthetic turbulent flow datasets generated via Direct Numerical Simulation (DNS) with varying geometries. Furthermore, the model and placement strategy were validated on a real-world sea surface temperature dataset (NOAA OISST V2), where they achieved a notable 14.01% improvement in Mean Squared Error (MSE) over the strongest competing baseline.
4. Clarity and Readability: The paper is well-written and structured logically, making the methodologies, including the intricacies of the GNN architecture and the PPO algorithm, as well as the extensive experimental results, clear and relatively easy for readers to follow.


Weaknesses
1. Limited Real-World Dataset Validation: While the performance on the single real-world sea surface temperature dataset is promising, the claim of general superiority on real-world data might be perceived as strong based on this sole example. Evaluating the model and placement strategy on additional real-world fluid dynamics datasets, potentially from different domains (e.g., aerodynamics, industrial flows) with varying complexities and sensor modalities, would more robustly confirm its broader applicability and effectiveness.
2. Absence of Ablation Studies: The paper could be enhanced by including detailed ablation studies for the key design choices within both the GNN and the PPO algorithm. For the directional transport-aware GNN, this could involve analyzing the impact of the explicit encoding of directionality and information transport, or the contribution of different components in its encoder-processor-decoder framework. For the Two-Step Constrained PPO, an ablation could quantify the benefits of the two-stage training procedure versus a single-stage approach, or the sensitivity to the choice of constrained sampling (Gumbel top-k ) and log probability estimation (saddle point approximation ). Such studies would offer deeper insights into why specific components are crucial for the observed performance.
3. No Comparison on Navier-Stokes-Based Datasets: The paper's focus is on realistic conditions, moving away from datasets based on idealized physics or known governing equations. However, many existing deep learning methods for flow reconstruction are benchmarked on datasets derived from, for example, solutions to the Navier-Stokes equations. Including a comparison on some of these established (even if idealized) benchmarks could provide a broader context for the proposed model's performance and highlight its advantages or trade-offs relative to a wider spectrum of existing approaches that do rely on such physical priors.

---

> ### Author Rebuttal · Authors · 2025-07-30
>
> We thank you for your insightful comments on improving our paper. We hope the following could address your concerns.
>
> ## Ablation Studies on DTA-GNN
>
> We conduct ablation studies in the following table. ABL_d means DTA-GNN without directional information in the message passing stage. Formally, the edge equation in Equation 2 becomes
> $$
> \\tilde{e}_{i,j}^{\\ell} \\leftarrow \\mathcal{T} \\bigl(\\tilde{v}_j^{\\ell-1} - \\tilde{v}_i^{\\ell-1}\\bigr)
> $$
> ABL_diff means DTA-GNN without the difference between neighboring latent states. We simply concatenate the neighboring latent states and multiply with the direction information. MeshGraphNets corresponds to removing both the directional information and difference between neighboring latent states.
>
> | | MeshGraphNets|ABL_d|ABL_diff|DTA-GNN|
> |-|-|-|-|---|
> |5% Uniform| 78.421 | 75.124| 63.162  | 59.308|
> |10% Uniform|74.549 | 59.064| 52.294  | 43.647|
> |20% Uniform | 32.596 | 30.368 | 30.660  | 28.173|
> |30% Uniform | 29.817 | 26.923 | 27.497  | 24.883|
> |5% Random | 103.126 | 155.466 | 101.878 | 97.076 |
> |10% Random | 62.337 | 73.355 | 60.536  | 56.224 |
> |20% Random | 48.519 | 49.210 | 48.622  | 44.748 |
> |30% Random | 31.806 | 32.389 | 30.943  | 29.803 |
>
> Based on these results, we draw three key conclusions:
>
> (1) Directional information plays a critical role when sensor density is low or when sensors are placed randomly, as some regions may lack sufficient sensor coverage.
>
> (2) Relying solely on the difference between neighboring latent states is suboptimal, as this approach does not capture edge attributes or directional cues essential for accurate information transfer.
>
> (3) Simply concatenating neighboring latent states leads to a moderate drop in performance, indicating that explicitly computing transported information is beneficial. Nevertheless, since message passing networks can still approximate difference operators, the performance degradation is less severe than when directional information is entirely removed.
>
> ---
>
> ## Ablation Studies on Two-Step Constrained PPO
> We report the ablation studies on Two-Step Constrained PPO in the following table. Penalized PPO refers to the first stage, where we use a penalized objective function to guide the model toward the constraints. During inference, we use Gumble Top-k to strictly enforce the equality constraints. Constrained PPO refers to the second stage, where we use Gumble Top-k for sampling and saddle point approximation for computing the log probability. Two-Step PPO without Saddlepoint Approx refers to the original Two-Step Constrained PPO, but removing saddle point approximation in the second stage. In the second stage, we use unconstrained log probability.
>
> |Shape| Penalized PPO| Constrained PPO| Two-Step PPO without Saddlepoint Approx |
> |-|-|-|-|
> |Sphere|3.270|19.280|3.178|
> |Ellipsoid|11.002|31.647|10.629|
> |Cylinder|9.490|55.014|9.575|
> |Airfoil|46.431|327.966|44.103|
>
> From these results, we draw several important conclusions:
>
> (1) As discussed in the main paper, initiating constrained training from a randomly initialized policy is overly restrictive and substantially limits the model’s performance.
>
> (2) Although the penalized training in the first stage helps guide the model toward a reasonable sensor placement strategy, it does not strictly enforce the equality constraint. Consequently, when combined with a constraint-compliant sampling strategy during inference, its performance degrades.
>
> (3) Accurately computing the log probability in the second-stage constrained training is essential. Without the saddle point approximation, as in the Two-Step PPO without Saddle Point Approximation, the model fails to outperform even the baseline of uniformly placed sensors.
>
> ---
>
> ## Experiments on Turbulence Data
> We compare our proposed DTA-GNN against three strong baselines on Kolmogorov Flow and Taylor-Green Vortex. Both datasets are generated via high-resolution numerical simulations using a pseudo-spectral solver governed by the incompressible Navier–Stokes equations. Kolmogorov Flow is driven by time‑dependent sinusoidal forcing at Re = 2000, while Taylor–Green Vortex starts from its analytic solution with added Gaussian noise and is simulated at Re = 1500.
>
> **Kolmogorov Flow**
> |Method|5% Random|10% Random|20% Random|30% Random|
> |-|-|-|-|-|
> |Ours|9.484| 8.537|4.510|3.443|
> |FCN |10.547|9.1557|5.109|4.690|
> |DiffusionPDE|11.214|9.458| 6.699|4.703|
> |MeshGraphNets|10.180|9.283| 5.271|4.137 |
>
> **Taylor Green Vortex**
> |Method|5% Random|10% Random|20% Random|30% Random|
> |-|-|-|-|-|
> |Ours|6.749|4.856|2.643|1.267|
> |FCN|8.430|6.354|3.590|2.825 |
> |DiffusionPDE|8.898|5.348|3.142|1.974|
> |MeshGraphNets|9.801|5.677|3.119|2.831|
>
> ---
>
> ## Additional Real World Validation
> Unfortunately, most existing CFD datasets and benchmark studies rely on numerical simulations with simplified assumptions and lack real-world observational data. For instance, [1, 3] are based on Reynolds-Averaged Navier–Stokes (RANS) simulations, [2] employs Large-Eddy Simulations (LES), and [4] utilizes a hybrid RANS–LES turbulence modeling approach.
> To further evaluate the practical applicability of our proposed method, we benchmark it on a real-world air quality dataset from the United States. Following the setup used in the sea surface temperature experiment, we compare our model against three strong baselines: FlowCompletionNetwork (FCN), DiffusionPDE, and MeshGraphNets. Results are reported in the table below. In this experiment, we assume that sensors are uniformly distributed over 10% of the available spatial locations.
>
> | Method | MSE |
> |-|-|
> |Ours | 1.488 |
> |FCN | 2.257 |
> |DiffusionPDE | 1.607 |
> |MeshGraphNets | 1.628 |
>
>
> Additionally, our Navier–Stokes-based benchmarks are derived from high-fidelity simulations, offering a more realistic approximation of real-world fluid dynamics compared to simplified models. Given that our model consistently achieves strong performance across both synthetic and real-world datasets, we are optimistic about its potential for real-world applications.
>
> ---
>
> ## Robustness to Noise
> To evaluate robustness to sensor noise, we inject Gaussian noise N(0, 0.025) into the observed sensor data in the Airfoil dataset. Note that the noise is injected into normalized data that follow a standard normal distribution N(0,1). We compare the performance of three strong baselines, FlowCompletionNetwork (FCN), DiffusionPDE, and MeshGraphNets, with our proposed DTA-GNN.
> | Method| 5% Random | 10% Random| 20% Random| 30% Random|
> |-|-|-|-|-|
> |Ours| 102.098|60.083| 48.394| 30.732|
> |FCN| 134.292|81.654| 68.733| 52.278|
> |DiffusionPDE|115.646| 73.280| 58.348| 32.270|
> |MeshGraphNets|114.209| 70.263| 56.293| 34.174|
>
> ---
>
> ## Computation Cost and Scalability of PPO
> The PPO framework operates as a single-step decision-making process, where the sensor placement configuration is determined once and remains fixed throughout the entire trajectory. As a result, the computational cost of executing the PPO policy is minimal, particularly for long trajectories. Given the substantial improvements in reconstruction accuracy achieved by the learned placement strategy, we consider the use of PPO to be a worthwhile and effective choice.
> The underlying model architecture is based on a message-passing graph neural network (GNN), which scales linearly with the number of nodes, making it suitable for large-scale spatial domains.
>
> ---
>
> ## Sensor Placement Constraints
> The sensor placements by PPO are restricted to the boundaries to reflect real world scenarios.
>
> ---
>
> ## Interpretability of DTA-GNN
> The interpretability of our proposed model is grounded in its principled design, which draws from established physical and algorithmic analogies. First, by mimicking the structure of the advection operator, we endow our processor with a clear directional semantics. Second, the processor can be seen as a learnable generalization of classical interpolation schemes. Here, directional information modulates neighbor contributions, and the update rule mirrors interpolation by aggregating weighted differences, augmented with self-contributions. Finally, unlike conventional message passing GNNs that rely on generic concatenation and MLPs, our model introduces an explicit notion of directionality, enhancing interpretability by aligning message flow with physically meaningful pathways.
>
> ---
>
> ## Interpretability of PPO
> Our sensor placement policy is guided by both the variability of the underlying fluid field and the performance of the reconstruction model. It prioritizes regions with high variability while also selecting sensor configurations that the model can accurately reconstruct. However, since the reconstruction model functions as a black box, the resulting placement decisions are not readily interpretable.
> We argue that the lack of interpretability is not a limitation of our approach, but rather a reflection of the inherently opaque nature of the problem itself.
>
> ---
>
> ## Fixed Sensor Proportion in PPO
> Empirically, increasing the sensor budget consistently improves the performance. This reflects an inherent trade-off in the problem formulation.  At present, the sensor budget is specified in advance, and the model is tasked with determining optimal sensor locations. An interesting futuer direction is to reverse this setup and ask the policy to determine the number of sensors required with a desired level of reconstruction accuracy.
>
> ---
>
> We hope this clears up any confusion and concerns, and if so, we would greatly appreciate your consideration in raising the score.
>
> [1] AirfRANS: High Fidelity Computational Fluid Dynamics Dataset for Approximating Reynolds-Averaged Navier–Stokes Solutions
>
> [2] WindsorML: High-Fidelity Computational Fluid Dynamics Dataset For Automotive Aerodynamics
>
> [3] DrivAerNet: A Parametric Car Dataset for Data-Driven Aerodynamic Design and Prediction
>
> [4] AhmedML: High-Fidelity Computational Fluid Dynamics Dataset for Incompressible, Low-Speed Bluff Body Aerodynamics

---

> > ### Comment · Reviewer_4m3m · 2025-08-05
> >
> > Thanks for the responses from the authors. Most of my concerns have been resolved. However, the computational cost remains unclear. Therefore, I keep my rating as Borderline accept.

---

> > > ### Author Response · Authors · 2025-08-05
> > >
> > > We thank the reviewer for the feedback. We would like to further clarify the computational cost.
> > >
> > > In training, our reconstruction model requires approximately 16.6 GPU-hours, while the PPO placement policy takes about 30 GPU-hours. Although the placement policy demands more time, both remain modest on a modern GPU workstation. These measurements were obtained on an NVIDIA RTX 4090; using more advanced hardware such as an NVIDIA H100 or H200 could yield significant speed-ups.
> > >
> > > During inference, the reconstruction model processes each frame in 66.31 ms, and the sensor placement policy determines placement in 94.14 ms. For the airfoil dataset, where each trajectory consists of 1,000 frames, the placement policy runs only once, whereas reconstructing all frames takes 66.31 seconds (which can be accelerated via batch processing when data are pre-collected or under less stringent real-time constraints). As a result, the placement policy adds just 0.14 percent to the overall compute time.

---

### Official Review · Reviewer_Qp1z · 2025-06-28

**Clarity:** 3
**Significance:** 2
**Originality:** 2
**Rating:** 3
**Confidence:** 2

**Summary:**

This paper focuses on CFD data reconstruction from partial data. Considering an underlying mesh-based discretization with some sensors on the mesh boundary capable
of capturing some properties (pressure, velocity), the goal is to inferre the density and velocity field on the full mesh. The paper comes with two distinct contributions.
The first one proposes an extended  GNN architecture with transport capabilities. Experiments comparing with several other SOAT architectures show that the proposed one achieves
best performance   in this specific context of mesh-based reconstruction from partial data. Then a second contribution focused on optimizing sensor placement given a certain sensor budget.
The solution proposed is based on an adaptation of the classical PPO RL algorithm. Results comparing to basic baselines show some improvements.

**Questions:**

- Motivate your choice for the PPO algorithm and why the QR-pivoting and SVD approach fail in your experiments. Also comment on the recent paper (likely published after the submission) Diff-SPORT.

- Can you comment on comparing a DTA-GNN and the same GNN without the DTA part (MeshGraphNet is probably close)?

- Appendix G: why changing the sensor location at each frame ? Would expect the sensor to be fixed.

**Ethical Concerns:**

["NO or VERY MINOR ethics concerns only"]

**Final Justification:**

Many concerns raised in my review have been adressed during the rebuttal.

**Limitations:**

Limitations could be more thoroughly discussed and included in the core paper (currently in the appendix)

**Quality:**

2

**Strengths And Weaknesses:**

Clarity: The paper is generally clear and well written. The exposition of the contributions is somehow not well balanced, the PPO approach with the associated sampling issue  taking a large amount of space while experimental validation is very slim.

Quality:
- Dataset: mesh sizes, topology. Some more elements and some visuals would be welcome. It's difficult
  to figure out how difficult the problem tested are.
- Not totally clear how the data set is split into a validation and training set.
- Experiments for the GNN:
  - Several of the NNs used for comparison have been modified to work with mesh. This is explained in the appendix. Should be mentioned in the core paper as this can make a significant divergent architecture from the one proposed in the original paper.
  - The central proposed adaptation to GNN architecture (DTA-GNN) is a inner product to compute d, but proved effective as demonstrated by the experiments. I miss in the experiments a comparison with a GNN without the transport aspect and one with it, but no other change, to measure more directly the gain that adding d brings.
- PPO:
  - RL algorithms like PPO are effective to learn action policies to curb a trajectory toward a high cumulated reward. Optimal sensor placement does not naturally fit into that category.  Approaches like simulation based inference sound more directly related to this inverse problem of sensor placement. Could you develop in the paper some rational about the path taken and the choice of PPO beyond a validation by experiment ?
   - Experimental results show some gains compared to uniform and random sampling, but there is no discussion or number giving the cost of the PPO based approach. Should at least be discussed in the weaknesses.
   - QR pivoting and SVD are giving worst result than the base uniform/random strategies (in appendix). Why ? It eventually affects the experimental results as the approach proposed is only compared against very basic schemes.



Significance and Originality: the paper shows several weaknesses first on motivating the approach chosen, then in the experiments. This affects the quality and potential impact of the contribution.

---

> ### Author Rebuttal · Authors · 2025-07-30
>
> We thank you for your insightful comments on improving our paper. We hope the following could address your concerns.
>
> ## Dataset Description
>
> The sphere dataset contains 100 trajectories, each with a trajectory length of 500. We vary the sphere radius to create different trajectories. The minimum mesh size is 1794, and the maximum mesh size is 74148.
> The ellipsoid dataset contains 200 trajectories, each with a trajectory length of 250. We vary the three semi-axes to create different trajectories. The minimum mesh size is 3314, and the maximum mesh size is 94544.
> The cylinder dataset contains 100 trajectories, each with a trajectory length of 500. We vary the height and radius to create different trajectories. The minimum mesh size is 3450, and the maximum mesh size is 76304.
> The airfoil dataset contains 50 trajectories, each with a trajectory length of 1000. We vary the maximum camber and maximum thickness to create different trajectories. The minimum mesh size is 2519, and the maximum mesh size is 53786.
>
> ----------
>
> ## Train, Valid, Test Split
>
> We use 80% of the trajectories for training, 10% for validation, and 10% for testing. This means that the testing data are always out of distribution in terms of geometry configurations and inlet velocity.
>
>
> ----------
>
> ## Description on Baseline Modification
> We thank the reviewer for the suggestion. As we are unable to modify the paper during the rebuttal stage, we will implement changes in our future versions.
>
> ----------
>
> ## Ablation Studies on DTA-GNN
>
> We conduct ablation studies in the following table. ABL_d means DTA-GNN without directional information in the message passing stage. Formally, the edge equation in Equation 2 becomes
> $$
> \\tilde{e}_{i,j}^{\\ell} \\leftarrow \\mathcal{T} \\bigl(\\tilde{v}_j^{\\ell-1} - \\tilde{v}_i^{\\ell-1}\\bigr)
> $$
> ABL_diff means DTA-GNN without the difference between neighboring latent states. We simply concatenate the neighboring latent states and multiply with the direction information. MeshGraphNets corresponds to removing both the directional information and difference between neighboring latent states.
>
> |   | MeshGraphNets | ABL_d   | ABL_diff | DTA-GNN |
> |-|-|-|-|---|
> | 5% Uniform  	| 78.421 | 75.124| 63.162  | 59.308 |
> | 10% Uniform| 74.549 | 59.064| 52.294  | 43.647 |
> | 20% Uniform | 32.596 | 30.368 | 30.660  | 28.173 |
> | 30% Uniform | 29.817 | 26.923 | 27.497  | 24.883 |
> | 5% Random | 103.126 | 155.466 | 101.878 | 97.076 |
> | 10% Random | 62.337 | 73.355 | 60.536  | 56.224 |
> | 20% Random | 48.519 | 49.210 | 48.622  | 44.748 |
> | 30% Random | 31.806 | 32.389 | 30.943  | 29.803 |
>
> Based on these results, we draw three key conclusions:
>
> (1) Directional information plays a critical role when sensor density is low or when sensors are placed randomly, as some regions may lack sufficient sensor coverage.
>
> (2) Relying solely on the difference between neighboring latent states is suboptimal, as this approach does not capture edge attributes or directional cues essential for accurate information transfer.
>
> (3) Simply concatenating neighboring latent states leads to a moderate drop in performance, indicating that explicitly computing transported information is beneficial. Nevertheless, since message passing networks can still approximate difference operators, the performance degradation is less severe than when directional information is entirely removed.
>
> ----------
>
> ## Appendix G Changing Sensor Location for Randomly Placed Sensors
> We change the sensor location for randomly placed sensors to estimate the expected MSE. We report the results for sensors fixed for the entire trajectory. For each trajectory, we randomly sample the sensor locations, and this sensor placement is fixed for the entire trajectory.
>
> | Method        | MSE   |
> |---------------|-------|
> | Uniform       | 3.092 |
> | Random (change at every frame)        | 4.110 |
> | Random (fixed for trajectory)        | 7.531 |
> | QR Pivoting   | 6.988 |
> | d-optimal     | 6.309 |
>
>
> ----------
>
>
> ## Rationale for Choosing PPO
> The PPO framework used in this work is not based on the Markov decision process. As described in Section 4 (Optimal Sensor Placement), we instead cast the problem as a policy learning and optimization task, where the objective is to learn a generalizable policy that maps an input mesh geometry to a sensor placement configuration. This configuration should minimize reconstruction error while satisfying a strict equality constraint. The problem is inherently single-step decision-making.
>
> We choose PPO over supervised regression for several reasons. Supervised regression would require differentiating through the binary placement mask, which is constrained to satisfy an equality condition. This makes direct optimization intractable, as the best-known gradient estimator [1] has a computational complexity of $O(k |V|^2)$. As discussed in the paper, this is prohibitive for large meshes. Moreover, supervised regression models may extrapolate poorly at test time when encountering geometries outside the training distribution. In contrast, PPO learns to act based on reward feedback and can generalize more robustly, provided sufficient exploration during training. Finally, in realistic scenarios where sensor data may contain noise, the policy gradient in PPO naturally averages over such noise, contributing to greater training stability.
>
> [3] also applies a **single step PPO for optimization tasks**. For example, one of its experiments employs a single step PPO to determine the angle of attack that maximizes mean lift coefficients.
>
>
> ----------
>
>
> ## Explanation for why QR-pivoting and SVD Fail
> Both QR pivoting and D-optimal assume that reconstruction relies on linear independence of measurements. For example, QR Pivoting maximizes the numerical rank of the selected columns in the feature matrix. However, Deep learning models are highly nonlinear, so the important sensors determined by QR pivoting and D-optimal may be uninformative for a neural network.
> These methods assume a basis matrix, assuming linear combinations of basis vectors reconstruct the full state. However, deep learning models do not operate in the same space. Its latent features or inductive biases may not align with the modes captured by the basis.
>
>
> ----------
>
>
> ## Computational Cost of PPO
> The PPO formulation in our approach involves a single-step decision, where the sensor placement configuration is determined once and used throughout the entire trajectory. As a result, the computational overhead of executing the PPO policy is minimal, particularly when the trajectory length is large.
> Moreover, given the substantial improvements in reconstruction accuracy achieved by the learned sensor placement policy, we believe that the use of PPO is well justified and offers a favorable trade-off between computational cost and performance.
>
>
> ----------
>
>
> ## Comment of Recent Papers
> [2] violates several key assumptions outlined in our introduction. The experiments are limited to 2D domains, and the datasets are generated from Navier–Stokes simulations. Additionally, the method assumes that sensors can be arbitrarily placed within the flow field without affecting the underlying fluid dynamics, a simplification that is physically unrealistic in many practical scenarios.
>
>
> ----------
>
> We hope this clears up any confusion and concerns, and if so, we would greatly appreciate your consideration in raising the score.
>
>
>
> [1] SIMPLE: A Gradient Estimator for k-Subset Sampling, ICLR 2023.
>
> [2] Diff-SPORT: Diffusion-based Sensor Placement Optimization and Reconstruction of Turbulent flows in urban environments
>
> [3] Single-step deep reinforcement learning for open-loop control of laminar and turbulent flows, Phys. Rev. Fluids 6, 053902 – Published 12 May, 2021

---

> > ### Author Response · Authors · 2025-08-05
> >
> > Dear Reviewer,
> >
> >
> > This is a kind reminder that we are approaching the end of the author-reviewer discussion, and we have not yet received your feedback on our response. We fully understand and appreciate your valuable time and commitment. However, we are eager to engage further with you to improve the quality of our work, to which we have devoted extensive effort. We sincerely hope that our responses have addressed your concerns. We would greatly appreciate your further feedback or questions.
> >
> >
> > Warm regards,
> >
> > Authors

---

> > > ### Comment · Reviewer_Qp1z · 2025-08-06
> > >
> > > Thank you for the detailled answer to my questions. I appreciate the abbation study that directly answer some of my concerns. Thanks also for justifying the PPO use. I also note that you provided cost (performance) numbers as answer to an other reviewer. That was also what I expected in order to compare the gain versus cost of PPO approach compared to the baseline.  I am upgrading my rating.

---

> > > > ### Author Response · Authors · 2025-08-07
> > > >
> > > > We thank the reviewer for their valuable suggestions and for raising the score. We will incorporate the feedback from our discussion into the revised version of the paper.

---

### Official Review · Reviewer_dAAG · 2025-07-02

**Clarity:** 2
**Significance:** 3
**Originality:** 3
**Rating:** 4
**Confidence:** 1

**Summary:**

This paper proposed a realistic flow field reconstruction method based on GNN and a two-step constrained PPO for policy learning. A directional transport-aware GNN is designed to explicitly encode both flow directionality and information transport. This method can also allow for arbitary sensor placement and 3D domains. Extensive experiments prove that the proposed method can achieve state-of-the-art performance.

**Questions:**

Actually, I am not familiar with this research field, can authors clarify more about the Proposition 3 part? And what additional techniques are proposed based on existing method. Even though authors have supplemented with an algorithm illustration in the main maniscript about Two-Step Constrained PPO, it is still little confusing to researchers who have limited knowledge in this domain.

**Ethical Concerns:**

["NO or VERY MINOR ethics concerns only"]

**Final Justification:**

Most of my concerns are tackled, but I would follow other reviewers because of low overlap with my expertise.

**Limitations:**

Please refer to the weakness and question parts.

**Quality:**

3

**Strengths And Weaknesses:**

Strengths:

1. The method releases lots of problem assumptions like arbitary sensor placement, which is more close to realistic flow scenes.

2. The proposed novelty of transport-aware GNN and PPO seems to be resonable.

3. The method is extensively evaluated and proved effectively in the experiments, outperforming existing methods.

Weakness:

1. The paper lacks adequate ablation studies for each designed novelty. Incorporating these experiments can further strengthen the paper's solid contributions. For example, for the Directional Transport-Aware GNN, Two Step Constrained PPO, authors can give the experimental results without these designs.

2. The runtime of the proposed method can be tested and compared with other learning-based methods, like DiffusionPDE.

3. Since the latest compared baselines in this work is DiffusionPDE (NeurIPS'24), which is a little out-of-date, could authors compare the model with more recent methods in 2025?

---

> ### Author Rebuttal · Authors · 2025-07-30
>
> We thank you for your insightful comments on improving our paper. We hope the following could address your concerns.
>
> ## Ablation Studies on DTA-GNN
>
> We conduct ablation studies in the following table. ABL_d means DTA-GNN without directional information in the message passing stage. Formally, the edge equation in Equation 2 becomes
> $$
> \\tilde{e}_{i,j}^{\\ell} \\leftarrow \\mathcal{T} \\bigl(\\tilde{v}_j^{\\ell-1} - \\tilde{v}_i^{\\ell-1}\\bigr)
> $$
> ABL_diff means DTA-GNN without the difference between neighboring latent states. We simply concatenate the neighboring latent states and multiply with the direction information. MeshGraphNets corresponds to removing both the directional information and difference between neighboring latent states.
>
> |   | MeshGraphNets | ABL_d   | ABL_diff | DTA-GNN |
> |-|-|-|-|---|
> | 5% Uniform  	| 78.421 | 75.124| 63.162  | 59.308 |
> | 10% Uniform| 74.549 | 59.064| 52.294  | 43.647 |
> | 20% Uniform | 32.596 | 30.368 | 30.660  | 28.173 |
> | 30% Uniform | 29.817 | 26.923 | 27.497  | 24.883 |
> | 5% Random | 103.126 | 155.466 | 101.878 | 97.076 |
> | 10% Random | 62.337 | 73.355 | 60.536  | 56.224 |
> | 20% Random | 48.519 | 49.210 | 48.622  | 44.748 |
> | 30% Random | 31.806 | 32.389 | 30.943  | 29.803 |
>
> Based on these results, we draw three key conclusions:
>
> (1) Directional information plays a critical role when sensor density is low or when sensors are placed randomly, as some regions may lack sufficient sensor coverage.
>
> (2) Relying solely on the difference between neighboring latent states is suboptimal, as this approach does not capture edge attributes or directional cues essential for accurate information transfer.
>
> (3) Simply concatenating neighboring latent states leads to a moderate drop in performance, indicating that explicitly computing transported information is beneficial. Nevertheless, since message passing networks can still approximate difference operators, the performance degradation is less severe than when directional information is entirely removed.
>
> ----------
>
> ## Ablation Studies on Two-Step Constrained PPO
> We report the ablation studies on Two-Step Constrained PPO in the following table. Penalized PPO refers to the first stage, where we use a penalized objective function to guide the model toward the constraints. During inference, we use Gumble Top-k to strictly enforce the equality constraints. Constrained PPO refers to the second stage, where we use Gumble Top-k for sampling and saddle point approximation for computing the log probability. Two-Step PPO without Saddlepoint Approx refers to the original Two-Step Constrained PPO, but removing saddle point approximation in the second stage. In the second stage, we use unconstrained log probability.
>
> | Shape| Penalized PPO| Constrained PPO| Two-Step PPO without Saddlepoint Approx |
> |-|-|-|-|
> | Sphere | 3.270 | 19.280| 3.178|
> | Ellipsoid | 11.002| 31.647| 10.629 |
> | Cylinder | 9.490| 55.014| 9.575|
> | Airfoil| 46.431| 327.966  | 44.103 |
>
> From these results, we draw several important conclusions:
>
> (1) As discussed in the main paper, initiating constrained training from a randomly initialized policy is overly restrictive and substantially limits the model’s performance.
>
> (2) Although the penalized training in the first stage helps guide the model toward a reasonable sensor placement strategy, it does not strictly enforce the equality constraint. Consequently, when combined with a constraint-compliant sampling strategy during inference, its performance degrades.
>
> (3) Accurately computing the log probability in the second-stage constrained training is essential. Without the saddle point approximation, as in the Two-Step PPO without Saddle Point Approximation, the model fails to outperform even the baseline of uniformly placed sensors.
>
> ----------
>
> ## Runtime and Memory Comparison
> We report the runtime and memory consumption of various methods in the table below. The results demonstrate that our proposed model is both memory-efficient and computationally fast. This is consistent with its parameter-efficient design, as it uses the fewest parameters among all compared models.
>
> | Method                | Memory (MB) | Runtime (ms) |
> |-----------------------|-------------|--------------|
> | Ours                  | 434.16      | 66.31        |
> | MeshGraphNets         | 482.40      | 70.50        |
> | Diffusion PDE         | 491.98      | 715.65       |
> | FlowCompletionNetwork | 584.65      | 64.16        |
> | OFormer               | 1584.08     | 81.14        |
> | GKO                   | 909.80      | 79.13        |
>
> ----------
>
> ## Additional Baseline Comparison
> We include [1] as our baseline and conduct experiments on the airfoil dataset. Results show that our DTA-GNN remains the best method.
>
> |        | MSE    |
> |---------------|----------|
> | 5% Uniform  | 117.122  |
> | 10% Uniform   | 75.871   |
> | 20% Uniform   | 49.478   |
> | 30% Uniform  | 30.119   |
> | 5% Random   | 130.775  |
> | 10% Random    | 64.194   |
> | 20% Random   | 51.404   |
> | 30% Random   | 34.317   |
>
> ----------
>
> ## Additional Clarification on Proposition 3 and Two-Step Constrained PPO
> Given the constrained problem defined in Equation 3, a natural approach is to directly apply constrained sampling(Section 4.1) and constrained log probability computation (Section 4.2), forming the basis of the Constrained Training Stage. However, as demonstrated in the ablation studies of the Two-Step Constrained PPO, enforcing the equality constraint from the outset is overly restrictive for a randomly initialized sensor placement policy, often leading to suboptimal solutions.
> This limitation motivates Proposition 3, which reformulates the constrained objective into a penalized objective function. Under this formulation, the policy is not required to satisfy the constraint exactly; instead, the objective function encourages the policy to produce outputs that approximately respect the constraint. Nevertheless, the ablation results indicate that this penalized formulation alone is insufficient for producing an effective placement policy for inference, as it lacks architectural mechanisms to enforce the constraint precisely.
>
> To address this, we propose the Two-Step Constrained PPO. In the first stage, we use Penalized Training to guide the policy toward constraint-compliant behavior, helping it to learn a meaningful initialization. In the second stage, we transition to Constrained Training, incorporating architectural components that strictly enforce the equality constraint. This two-stage procedure enables the model to achieve both feasibility and optimality in sensor placement.
>
>
> ----------
>
> We hope this clears up any confusion and concerns, and if so, we would greatly appreciate your consideration in raising the score.
>
>
> [1] Qitian Wu and Chenxiao Yang and Kaipeng Zeng and Michael Bronstein, Supercharging Graph Transformers with Advective Diffusion, ICML 2025

---

> > ### Comment · Reviewer_dAAG · 2025-08-04
> > **Final comment**
> >
> > Thanks very much for authors' effort in rebuttal. Most of my concerns have been tackled. So I tend to remain the acceptance review.
> >
> > However, I strongly suggest authors should add these ablation studies and comparisons for better understanding of each novelty in the final version if accepted.

---

> > > ### Author Response · Authors · 2025-08-05
> > >
> > > We thank the reviewer for the valuable suggestions and recommendations for acceptance. We will include the ablation studies and additional comparisons in the final version.

---

### Official Review · Reviewer_5qAz · 2025-07-02

**Clarity:** 3
**Significance:** 3
**Originality:** 3
**Rating:** 4
**Confidence:** 4

**Summary:**

The authors proposed a directional transport-awareGraph Neural Network (GNN) that explicitly encodes both flow directionality and
information transport. The authors also studies the problem of optimal placement of sensors for enhanced reconstruction accuracy. They proposed a two-step constrained proximal policy optimisation based algorithm for that purpose.

**Questions:**

1. See Weaknesses 1-4.
2. Do you assume the random variables $a_i,~i \in \{1,2,\cdots,|V|\}$ to be mutually independent? If yes, does this assumption make sense when a_i's are observables at nodes in the graph, and usually they are correlated?

**Ethical Concerns:**

["NO or VERY MINOR ethics concerns only"]

**Final Justification:**

Most of my concerns are resolved.

**Limitations:**

yes

**Quality:**

3

**Strengths And Weaknesses:**

Strengths:

1. The proposed method offers significant performance improvements over the related baselines across various datasets of differing geometries.
2. The proposed algorithm offers improvement over existing sensor placement strategies.

Weaknesses:

1. **Novelty of direction information:** Direction information has been incorporated in [2] for GNNs on fluid reconstruction problem, so what are the differences with [2] and the authors directional transport aware GNN?
2. **Lacking experiments on turbulence data:** The paper is lacking experiments on turbulent flow. Can the authors provide experimental results on datasets similar to the Forced or Decayed turbulence as in [6,8]?
3. **Lacking Generalisability experiments:** The paper is lacking study on how the trained model generalises to unknown physical conditions as in [9]?
4. **Insufficient analysis:** It is important to assess if the reconstructed flow field accurately predicts Force coefficients e.g. lift and drag coefficients for practical applications as done in existing works [1,2, Zhong et al. 2023 in the paper]. Have the authors analysed the boundary layer profiles of the reconstructed flow filed on the datasets?

5. Related works are quite light, missing many important recent papers: [1-9]

[1] AirfRANS: High Fidelity Computational Fluid Dynamics Dataset for Approximating Reynolds-Averaged Navier–Stokes Solutions, NeurIPS 2022.

[2] Finite Volume Features, Global Geometry Representations, and Residual Training for Deep Learning-based CFD Simulation, ICML 2024.

[3] Topology-aware Neural Flux Prediction Guided by Physics, ICML 2025.

[4] PDE-Transformer: Efficient and Versatile Transformers for Physics Simulations, ICML 2025.

[5] PhyMPGN: Physics-encoded Message Passing Graph Network for spatiotemporal PDE systems, ICLR 2025.

[6] Multi-Order Loss Functions For Accelerating Unsteady Flow Simulations with Physics-Based AI, CAI 2024.

[7] Learning to simulate complex physics with graph networks, ICML 2020

[8] Machine learning–accelerated computational fluid dynamics, PNAS 2021

[9] Combining Differentiable PDE Solvers and Graph Neural Networks for Fluid Flow Prediction, ICML 2020.

---

> ### Author Rebuttal · Authors · 2025-07-30
>
> We thank you for your insightful comments on improving our paper. We hope the following could address your concerns.
>
> ## Novelty of Direction Information
>
> The Directional Integrated Distance (DID) proposed in [1] is fundamentally different from the approach presented in our work. Below, we highlight the key distinctions across several dimensions:
>
> 1. **Problem Definition**
> The work in [1] addresses the task of predicting velocity and pressure fields for steady-state simulations. Their model takes mesh coordinates as input and directly outputs steady-state velocity and pressure. In contrast, our work focuses on flow field reconstruction, where the input includes both the mesh and partially observed velocity and pressure data. The goal is to reconstruct the velocity and pressure at unobserved mesh points. Thus, our model must learn not only the underlying physical dynamics but also how to effectively integrate available observations for accurate reconstruction.
> Moreover, the simulations in [1] assume a constant inlet velocity, leading the solution to converge to a steady state. In our case, the inlet velocity is time-dependent and stochastic, making it infeasible to infer the full flow field from mesh geometry alone.
>
> 2. **Semantic Meaning of DID vs. Our Directional Information**
> In [1], the authors note that mesh coordinates provide limited information about immersed objects and their surroundings. To address this, they propose DID, which augments mesh coordinates by encoding the average distance to domain boundaries over various angular sectors.
> In contrast, our directional information encodes the alignment of a node with the local direction of information transfer. This metric is influenced not only by mesh geometry but also by observed sensor data.
>
> 3. **Role Within the Model**
> The DID in [1] is a precomputed geometric feature derived from mesh coordinates and is used as an additional input to their neural network. Our directional information, by comparison, is computed dynamically during message passing using a node's latent representation and edge features. It plays a functional role in guiding the aggregation of information.
>
>
> ----------
>
>
> ## Experiments on Turbulence Data
> We compare our proposed DTA-GNN against three strong baselines, FlowCompletionNetwork (FCN), DiffusionPDE, and MeshGraphNets, on two benchmark datasets: Kolmogorov Flow and Taylor-Green Vortex. Both datasets are generated via high-resolution numerical simulations using a pseudo-spectral solver governed by the incompressible Navier–Stokes equations.
> The Kolmogorov Flow dataset features a time-dependent sinusoidal external forcing and is simulated at a Reynolds number of 2000. The Taylor-Green Vortex dataset is initialized from its analytical solution and perturbed with Gaussian noise to produce a variety of flow trajectories. Simulations are carried out at a Reynolds number of 1500. The performance of each method on these datasets is summarized in the table below:
>
> **Kolmogorov Flow**
> | Method| 5% Random | 10% Random | 20% Random | 30% Random |
> |-|-|-|-|-|
> | Ours| 9.484    | 8.537| 4.510| 3.443|
> | FCN | 10.547   | 9.1557| 5.109| 4.690|
> | DiffusionPDE | 11.214 | 9.458| 6.699| 4.703|
> | MeshGraphNets | 10.180 | 9.283| 5.271| 4.137 |
>
> **Taylor Green Vortex**
> | Method | 5% Random | 10% Random | 20% Random | 30% Random |
> |-|-|-|-|-|
> | Ours | 6.749    | 4.856  | 2.643 | 1.267 |
> | FCN | 8.430    | 6.354  | 3.590 | 2.825  |
> | DiffusionPDE | 8.898 | 5.348 | 3.142   | 1.974 |
> | MeshGraphNets| 9.801 | 5.677 | 3.119  | 2.831|
>
>
> ----------
>
>
> ## Generalisability Experiments
> We evaluate the generalization capability of our proposed DTA-GNN by training it on the Ellipsoid dataset and testing it on the Sphere dataset. Notably, the Ellipsoid dataset contains no sphere geometries, ensuring that the test domain represents a previously unseen configuration. Furthermore, the two datasets differ in their inlet velocity profiles, resulting in entirely distinct flow fields.
> Despite these differences, as shown in the table below, DTA-GNN demonstrates strong generalization performance under these shifted conditions. In all test scenarios, except for the 30% Random setting (x% Random refers to sensors randomly distributed at x% of the mesh points), DTA-GNN consistently outperforms all baseline models, maintaining superior accuracy on the unseen flow distributions.
>
> |      | MSE for model trained on this dataset | Generalization MSE | % Drop in Performance |
> |-------------|-|----------|----------------|
> | 5% Random |7.180 | 7.650 | 6.54%  |
> | 10% Random | 4.110 | 4.468 | 8.71%  |
> | 20% Random | 2.155 | 2.328 | 8.03% |
> | 30% Random| 1.446 | 1.574| 8.86% |
>
>
>
> ----------
>
>
> ## Analysis on Lift and Drag Coefficients
> We report the MSE on lift and drag coefficients. The experiments are conducted on the airfoil dataset with 10% uniformly placed sensors.
>
> | Method | lift mse| drag mse |
> |-|-|-|
> | Ours| 2.54e-06 | 3.42e-07 |
> | MeshGraphNets | 7.43e-06  | 1.15e-06 |
> | DiffusionPDE| 4.47e-05 | 2.17e-06  |
> | FlowCompletionNetwork | 8.27e-06| 5.37e-06 |
>
>
> As shown in the results, our proposed model achieves the lowest error, indicating superior performance compared to baseline methods. This suggests that the model not only captures local flow patterns but also effectively reconstructs the global structure of the flow field. Moreover, the accuracy on lift and drag coefficients highlights the practical utility of our approach for downstream tasks such as aerodynamic analysis and design optimization.
>
>
> ----------
>
>
> ## Related Work
> We thank the reviewer for highlighting these related works. When preparing the manuscript, our primary focus was on studies concerning flow field reconstruction, as this is the central theme of our work. However, we agree that some papers on physical simulation and neural PDE solvers are also relevant to our approach. As we are unable to revise the manuscript at this stage, we will ensure that these works are included and properly discussed in future revisions.
>
>
> ----------
>
>
> ## Independence Assumption of Random Variables
> We assume that the random variables are independent. This assumption is reasonable because each sensor provides measurements only at its specific location, and the removal of one sensor does not directly influence the others. However, when the total number of sensors is constrained, statistical correlation is introduced through the equality constraint.
>
>
> ----------
>
>
> We hope this clears up any confusion and concerns, and if so, we would greatly appreciate your consideration in raising the score.
>
> [1] Finite Volume Features, Global Geometry Representations, and Residual Training for Deep Learning-based CFD Simulation, ICML 2024.

---

> > ### Comment · Reviewer_5qAz · 2025-08-03
> >
> > Most of my concerns have been resolved, so I have raised my score.
> >
> > I suggest that the author incorporate the additional experimental results and distinction with existing works regarding directional information preservation into the paper to clarify the significance of this work better.
> >
> > I do not see any results regarding the boundary layer profiles of the reconstructed flow field on the datasets. Although the Lift and drag coefficient results have been provided, I recommend that the authors supplement this with boundary layer profile analysis to make a more convincing case for this work.

---

> > > ### Author Response · Authors · 2025-08-05
> > >
> > > We thank the reviewer for the valuable suggestions and for raising the score. We will include these additional results, clarify distinctions with existing works in the camera-ready version, and continue exploring boundary-layer profile analysis.

---

### Note · Authors · 2025-08-12

We thank the reviewers for their constructive feedback and discussions, and the AC and SAC for coordinating the rebuttal phase. We are delighted by the supportive reception of our submission, with all reviewers giving positive scores. Reviewers highlighted the novelty of our DTA-GNN and Two-Step Constrained PPO for optimal sensor placement, as well as the significant performance improvements over related baselines across comprehensive experiments.

Inspired by their feedback, we clarified: (i) We conducted extensive ablation studies to isolate the individual contribution of each module in DTA-GNN and Two-Step Constrained PPO. (ii) We conducted additional experiments on turbulent flow data generated through high-fidelity Navier–Stokes simulations. (iii) We provided a comprehensive analysis of our model’s generalization, downstream applications for lift and drag coefficient calculation, and robustness to noise. (iv) We clarified the training and inference costs of Two-Step Constrained PPO, showing that its performance gain incurs only a 0.14% increase in overall inference time. (v) We justified why PPO is used for sensor placement.

We are happy that many reviewers noted the significance of our contributions and increased their support. Our work introduces a realistic problem formulation for fluid-field reconstruction and generates extensive datasets that closely mimic real-world scenarios. We develop a directional transport-aware GNN that explicitly encodes both directionality and information transport. We find that conventional sensor placement algorithms fail to identify effective sensor locations; thus, we propose a novel Two-Step Constrained PPO training strategy to learn a policy that identifies the optimal placement of sensors. Our reconstruction framework and optimal sensor placement strategy offer transformative potential across various scientific domains. We are confident in their real-world applications based on our comprehensive evaluations. Upon acceptance, we will make the promised changes and release all code and datasets for reproducing the results in this paper.

---

### Decision · Program_Chairs · 2025-09-17

**Decision:**

Accept (poster)

**Comment:**

This paper introduces a Directional Transport-Aware GNN for flow reconstruction and a Two-Step Constrained PPO for sensor placement under realistic constraints. While sensor placement is a long-standing optimization problem, the contribution here is in adapting PPO with equality constraints and coupling it with a reconstruction model under a realistic CFD setting.

Reviewers initially raised concerns about novelty (directional encoding overlap), limited turbulence experiments, and missing ablations. The rebuttal provided careful clarifications and additional results, largely addressing these points.

The AC finds the work a solid contribution: the originality is somewhat incremental, but the problem formulation is well-motivated, the empirical validation is comprehensive, and the paper illustrates the broader value of applying ML methods to challenging scientific domains.